



# Climate and Cryosphere Cause Regime Shifts in Water
# Yield over the Upper Brahmaputra River
Hao Li[1], Liu Liu[2], Baoying Shan[3], Lei Wang[4], Akash Koppa[1], Feng Zhong[5], Dongfeng Li[6],
Xuanxuan Wang[2], Wenfeng Liu[2], Xiuping Li[4], and Zongxue Xu[7]
[1]Hydro-Climate Extremes Lab, Ghent University, Ghent, Belgium
[2]Center for Agricultural Water Research in China, China Agricultural University, Beijing, China
[3]Research Unit Knowledge-based Systems, Ghent University, Ghent, Belgium
[4]Institute of Tibetan Plateau Research, Chinese Academy of China, Beijing, China
[5]College of Hydrology and Water Resources, Hohai University, Nanjing, China
[6]Department of Geography, National University of Singapore, Singapore
[7]College of Water Sciences, Beijing Normal University, Beijing, China
*Correspondence to*: Liu Liu (liuliu@cau.edu.cn)





**Abstract.** Although evidence of hydrological responses to climate is abundant, changes in water yield (WY) in mountainous regions due to climate change and intensified cryospheric melt remain unclear, mainly because of limited observations and large uncertainties in cryosphere-hydrological modeling. In this study, we used annual runoff observations and a high-resolution precipitation dataset to examine the long-term changes in WY in the Upper Brahmaputra River (UBR) basin, as represented by six sub-basins from the stream head to downstream. We found that WY generally increased during 1982–2013, but regime shifts were detected in the late 1990s. Moreover, the direction of the changes in WY reversed from increasing to decreasing in recent years despite the magnitude of the changes continually increasing from less than 10% to 80.5%. Furthermore, we used the double mass curve technique to assess the effects of climate, vegetation, and the cryosphere on WY. The results showed that the climate and cryosphere together contributed to over 80% of the magnitude increases in WY over the entire UBR basin. However, the combined effects were either offsetting or additive, further leading to slight or substantial magnitude increases, respectively, in which the role of vegetation was nearly negligible. Nevertheless, we found that meltwater from the cryosphere had the potential to alleviate the loss of water availability, which mainly resulted from reduced effective precipitation in most regions. Therefore, the combined effects of climate and cryosphere changes should be considered in ecological restoration and water resources management, particularly involving co-benefits for upstream and downstream regions.



**1 Introduction**

Water yield (WY) in mountains is crucial for sustaining fragile ecosystems in the headwaters, supplying valuable freshwater resources to downstream lowlands, and balancing co-benefits between the upstream and downstream areas, especially for large transboundary river systems (Viviroli et al., 2011). In mountainous regions, changes in WY have been commonly, but separately, attributed to climate changes (Dierauer et al., 2018; Song et al., 2021), vegetation (Goulden and Bales, 2014; Zhou et al., 2021), and the cryosphere (such as glacial snow melt; see Kraaijenbrink et al. 2021). These changes are expected to alter the spatial and temporal distribution of water resources (Tang et al., 2019) and further threaten the water supply and food security downstream (Biemans et al., 2019). Despite some in situ observations and runoff estimates from state-of-the-art remote sensing technology, the total river runoff for the Third Pole, which is also known as the "Asian Water Tower," has never been reliably quantified, and its responses to climate change remain unclear (Wang L et al., 2021). Therefore, comprehensively assessing the impacts of the climate, vegetation, and cryosphere on long-term changes, particularly in magnitude and direction, in WY in this region is of great importance for the sustainable development of water resources and ecological environment (Yao et al., 2019).

The Qinghai-Tibet Plateau (QTP), regarded as the center of the Third Pole, is one of the most sensitive and vulnerable mountainous regions to environmental changes (Kang et al., 2010; Yao et al., 2010, 2019) and supplies water resources for major rivers in Asia, such as Brahmaputra, Salween, Mekong, Yangtze, Yellow, and Indus Rivers. Changes in WY in this region are a crucial factor in the use of water resources, prevention of natural disasters, and protection of aquatic functions for the livelihoods of approximately two billion people in the area (Immerzeel et al., 2010). In recent years, changes in the climate, vegetation, and cryosphere have significantly affected the WY over the QTP (Bibi et al., 2018). For example, Fan and He (2015) highlighted the effects of precipitation on the direction of change in WY over the Salween and Mekong River basins. Li et al. (2020) determined that elevated precipitation and warming-induced changes in glacial snow patterns both contributed to the magnitude of the increase in WY for the Tuotuo River (a headwater of the Yangtze River). Similarly, Lutz et al. (2014) projected that increased precipitation near the Salween and Mekong Rivers and accelerated meltwater near the Indus River caused major changes in WY. Moreover, the role of vegetation in mountain water resources is important. Li et al. (2017) showed that increased evapotranspiration, mostly due to grassland restoration, decreased the WY in the Yangtze River basin, while Li et al. (2021) suggested that vegetation greening was mainly linked to the positive WY trend during the dry season over the Brahmaputra River.

Although a growing body of evidence has shown that WY is affected by climate, vegetation, and cryosphere in the QTP, most studies have focused on individual sub-basins and have not considered these three aspects together throughout this large and understudied region(Dierauer et al., 2018; Goulden and Bales, 2014; Kraaijenbrink et al. 2021; Song et al., 2021; Zhou et al., 2021). Therefore, previous results may not fully reveal the spatial variability in the region. Of specific interest is the Upper Brahmaputra River (UBR) basin, which covers an area of over 198,636 km$^2$ (Table S1) and has large gradients in elevation, climate, and vegetation (Li et al., 2019b). Therefore, providing a comprehensive, spatially differentiated study of the WY changes in the UBR basin that considers the joint effects of the climate, vegetation, and cryosphere is imperative. However, studies of WY changes in this region are significantly





hindered by the sparse network of hydrological observation stations (Li et al., 2019b; Wang L et al., 2021;
Yao et al., 2019), which leads to large uncertainties in WY forecasts and, thus, water resources assessments.
In addition, current precipitation estimates are highly uncertain owing to the complex topography of the
region, which limits the ability to accurately model the relationships between precipitation and runoff (Sun
and Su, 2020). Lastly, the present limited understanding of WY responses to the joint interaction of the
climate, vegetation, and cryosphere has become the biggest challenge for developing accurate physically-
based cryosphere-hydrological models (Pellicciotti et al., 2012). Nevertheless, long-term runoff data and
high-resolution satellite records of climate and vegetation cover provide a potential pathway for
determining their relationships using statistical methods.

In this study, we collected annual runoff data for 1982–2013 from six hydrological stations to detect

long-term changes in the WY over the UBR basin. In addition, a modified double mass curve (DMC)
method was implemented to assess the influence of climate, vegetation, and the cryosphere on WY.
Accordingly, the main objectives of this study were to identify the magnitude and direction of changes in
WY based on observed runoff data and quantify the contributions of the climate, vegetation, and
cryosphere to these changes. This study can provide a reference for physical-based cryosphere-
hydrological modeling and important information for water resources and ecosystem management over the
UBR basin and other mountainous regions.

## 2 Data and Methods

### 2.1 Study area

The Brahmaputra River (known as the Yarlung Zangbo River, or YZR, in China), a transboundary river in
the southern QTP, originates in the Gyama Langdzom Glacier and flows across China, India, and
Bangladesh, before emptying into the Indian Ocean. The UBR basin is located above the Nuxia
hydrological station (Fig. 1a), and its flow has significant implications for the ecology of the source region
and freshwater resources of South Asia. Here, we divided the UBR basin into the headstream (HYZR),
upstream (UYZR), midstream (MYZR), downstream (LYZR), Nianchu River (NCR), and Lhasa River
(LSR) by hydrological stations (Fig. 1b and Table S1), and analyzed WY changes over these six sub-basins
to reveal spatial differences.

The elevation gradient and the distance to the ocean in the UBR basin together contribute to a large

spatial variability in the climate (Sang et al., 2016; Wang Y et al., 2020, 2021). The annual precipitation in
the HYZR basin is less than 400 mm, while that in the LYZR basin is nearly 1000 mm (Fig. S1). Similarly,
the annual actual evapotranspiration (AET) increases gradually from upstream to downstream areas (Fig.
S1). Meanwhile, water and energy availability modulate the vegetation conditions (Li et al., 2019a);
vegetation cover increases dramatically from the HYZR to the LYZR basin (Fig. S1). Furthermore, glacial
snow meltwater from the cryosphere due to warming conditions has substantially affected the hydrology of
this region (Cuo et al., 2019; Yao et al., 2010; Wang L et al., 2021).



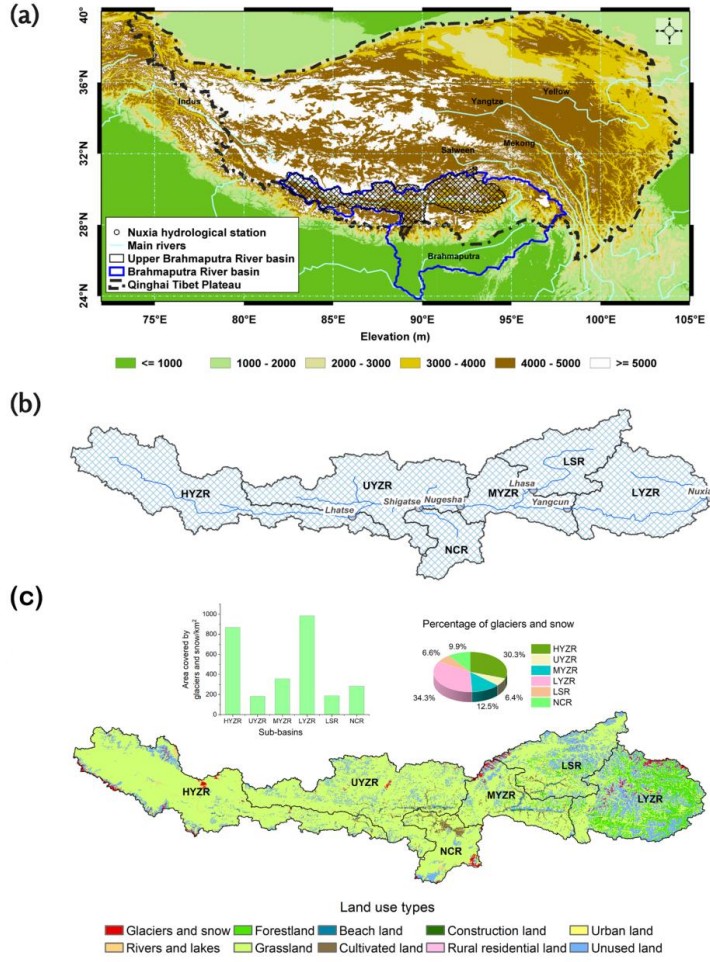

**Figure 1.** Location of **(a)** the Upper Brahmaputra River (UBR) basin over the Qinghai Tibet Plateau; **(b)** the six sub-basins delineated by the Lhatse, Nugesha, Shigatse, Yangcun, Lhasa, and Nuxia hydrological stations; and **(c)** the distribution of land use types and percentage of area covered by glaciers and snow in 2015, provided by National Tibetan Plateau Data Center (http://data.tpdc.ac.cn).

**2.2 Dataset**

**2.2.1 Runoff data**

Annual runoff data between 1982 and 2013 from six hydrological stations along the mainstream and major branches, which were provided by the Hydrology and Water Resources Survey Bureau of the Tibet Autonomous Region, were used in the study. The WY in the HYZR was determined by the runoff observed at the Lhatse hydrological station, while the WY in other sub-basins was determined by the difference between runoff observed from gauging stations located at the downstream station and that at the upstream and branch stations. For example, WY in the MYZR basin was equal to the difference between the observed annual runoff in the Yangcun hydrological station and that in the Lhasa and Nugesha stations





(Fig. 1b).

**2.2.2 Climate data**

The most recent 10 km gridded daily precipitation dataset was obtained from Sun and Su (2020), which
combined topographic and linear correction approaches based on 262 rain-gauge observations, and was
applied to estimate regional annual precipitation (P) in this study. Regional annual AET was acquired from
the Global Land Evaporation Amsterdam Model (GLEAM) products with a spatial resolution of 0.25°
(Martens et al., 2017). The effective precipitation (eP) was regarded as a proxy for climate in this study and
was calculated as the difference between P and AET, as shown in Section 2.3.2.

**2.2.3 Vegetation data**

The leaf area index (LAI) data used in this study were obtained from the Global Inventory Monitoring and
Modelling System (GIMMS) (https://ecocast.arc.nasa.gov/data/pub/gimms), and spanned 1982 to 2015
with a spatial resolution of 8 km × 8 km. GIMMS LAI3g (Zhu et al., 2013) was generated using an
artificial neural network trained on the Collection Terra Moderate Resolution Imaging Spectroradiometer
(MODIS) LAI product and the latest version of GIMMS NDVI3g (normalized difference vegetation index)
data for the same period, which has been proven to have an improved multi-sensor record harmonization
scheme compared to other global LAI products (Forzieri et al., 2020; Gonsamo et al., 2021). Note that all
gridded data were aggregated to regional values over each sub-basin on an annual time scale from 1982 to
2013, considering area-weighted effects.

**2.3 Methodology**

**2.3.1 Trend and abruption analysis**

In this study, we used the non-parametric Mann–Kendall test (Kendall, 1938; Mann, 1945) to identify the
trends in WY, and the non-parametric Pettitt abrupt detection method (Pettitt, 1979) to identify the turning
points (TP) in WY. The level of significance was set at 0.05. We compared the average WY before and
after each TP to reflect the magnitude of WY changes, and compared the trends before and after each TP to
reflect the direction of the changes.

**2.3.2 Double mass curve**

In a large and pristine mountainous river basin with diverse vegetation, climatic variability, cryospheric
melt, and vegetation dynamics are the three primary drivers of hydrological variation. Climatic variability
is typically more dominant and can often obscure the effects of other changes on hydrology (Cong et al.,
2009). The climatic effects on the annual WY must be excluded to enable quantification of the relative
contributions of the cryosphere and vegetation. According to the river basin water balance, the WY is
determined by the difference between precipitation, evapotranspiration, and changes in soil water storage.
Annual changes in soil water storage can generally be assumed to be constant and minor terms in the water
balance equation (Wei et al., 2009; Zhang et al., 2001); therefore, WY is mainly affected by precipitation
and evapotranspiration. Furthermore, precipitation has been proven to be the dominant factor for runoff
variation in the UBR basin (Li et al., 2019b; Wang Y et a., 2021; Xin et al., 2021). Hence, we defined the




difference between precipitation and evapotranspiration as eP for WY, which was used as an integrated
index for climatic variability in this study.

Unlike the traditional DMC method, where the accumulated WY from the disturbed watershed is

plotted against the accumulated WY from an undisturbed watershed, the modified DMC plots accumulated
annual WY versus accumulated annual eP in the URB basin. Specifically, the modified DMC used in this
study is a plot of the cumulative data of one variable versus the cumulative data of another related variable
in a concurrent period. It has previously been used to assess the effects of climate (Gao et al., 2011), forest
disturbance (Wei and Zhang, 2010), wildfire (Hallema et al., 2018), and the cryosphere (Brahney et al.,
2017) on water resources. Here, we built two types of DMC plots to assess the effects of climate (eP),
vegetation (LAI), and the cryosphere on WY changes over the entire UBR basin (which are shown in Fig.
S2).

First, the inter-annual total WY deviation ($\Delta WY(t)$, black diamond in Fig. S2) can be calculated as the

difference between WY after a TP ($WY(t)$) and the average WY before that TP ( $\frac{\sum_{t=1}^{t=tp} WY(t)}{tp}$ ), as follows:

$$\Delta WY(t) = WY(t) - \frac{\sum_{t=1}^{t=tp} WY(t)}{tp}, t = tp+1, tp+2, ..., 32 \qquad (1)$$

Second, the regression equation between the cumulative eP ( $\sum eP$ ) and cumulative WY ( $\sum WY$ )

before the TP can be constructed as follows:

$$\sum WY = a_1 \sum eP + b_1 \qquad (2)$$

Similarly, the regression equation between the cumulative LAI ( $\sum LAI$ ) and cumulative WY ( $\sum WY$ )

before the TP can be constructed as follows:

$$\sum WY = a_2 \sum LAI + b_2 \qquad (3)$$

Third, WY changes caused by climate change ( $WY_c(t)$ ) can be calculated by inputting the cumulative

eP after the TP into Eq. 2. Therefore, the WY deviation caused by climate change ( $\Delta WY_c(t)$, blue bar in
Fig. S2) can be calculated as follows:

$$\Delta WY_c(t) = WY_c(t) - \frac{\sum_{t=1}^{t=tp} WY(t)}{tp}, t = tp+1, tp+2, ..., 32 \qquad (4)$$

Similarly, the WY changes caused by vegetation ( $WY_v(t)$ ) were calculated using Eq. 3, and the WY

deviation caused by vegetation ( $WY_v$, tan bar in Fig. S2) can be calculated as follows:

$$\Delta WY_v(t) = WY_v(t) - \frac{\sum_{t=1}^{t=tp} WY(t)}{tp}, t = tp+1, tp+2, ..., 32 \qquad (5)$$

Finally, the WY deviation caused by the cryosphere ( $\Delta WY_s$, red bar in Fig. S2) can be calculated as:

$$\Delta WY_s(t) = \Delta WY(t) - \Delta WY_c(t) - \Delta WY_v(t) \qquad (6)$$



**2.3.3 Attribution analysis on changes in water yield**
The average effects of climate, vegetation, and cryosphere on the magnitude of the changes in WY were
calculated as follows:

$$\overline{\Delta WY_c} = \frac{\sum_{t=t+1}^{t=32} WY_c(t)}{32 - tp}$$


$$\overline{\Delta WY_v} = \frac{\sum_{t=t+1}^{t=32} WY_v(t)}{32 - tp} \qquad (7)$$


$$\overline{\Delta WY_s} = \frac{\sum_{t=t+1}^{t=32} WY_s(t)}{32 - tp}$$


The relative contribution ($RC$), ranging from 0 to 100, of climate, vegetation, and cryosphere changes
on the magnitude can be calculated as follows:

$$RC_c = \frac{\overline{\Delta WY_c}}{\left|\overline{\Delta WY_c}\right| + \left|\overline{\Delta WY_v}\right| + \left|\overline{\Delta WY_s}\right|}$$


$$RC_v = \frac{\overline{\Delta WY_v}}{\left|\overline{\Delta WY_c}\right| + \left|\overline{\Delta WY_v}\right| + \left|\overline{\Delta WY_s}\right|} \qquad (8)$$


$$RC_s = \frac{\overline{\Delta WY_s}}{\left|\overline{\Delta WY_c}\right| + \left|\overline{\Delta WY_v}\right| + \left|\overline{\Delta WY_s}\right|}$$


In addition, we used the Pearson correlation coefficient ($r$) to quantify the relationships between the
total WY deviation ($\Delta WY(t)$) and its components, which were the WY deviation caused by climate
($\Delta WY_c(t)$), vegetation ($\Delta WY_v(t)$), and cryosphere ($\Delta WY_s(t)$). The Student's t-test was used to detect
statistical significance for the Pearson's correlation coefficient at a level of 0.05.
**3 Results**
**3.1 Long-term changes in historical water yield**
The detection of long-term changes in WY from 1982 to 2013 over the entire UBR basin is illustrated in
Fig. 2. We found that there was great spatial variability in the annual WY (Fig. 2a). The mean annual WY
was highest in the LYZR basin (over 600 mm), followed by that over the LSR basin (nearly 400 mm).
However, the mean annual WY in the HYZR and NCR basins was less than 100 mm. The spatial
variability in annual WY was consistent with that of precipitation (Fig. S3), which was mainly determined
by elevation and distance to the ocean (Sang et al., 2016). WY generally increased during the study period,
as shown by the positive slope in Fig. 2b, which is in agreement with previous studies on a single basin (Li
et al., 2021; Lin et al., 2020; Zhang et al., 2011). However, a significant trend was only detected in the
UYZR and MYZR basins (hatched areas in Fig. 2b) in this study.

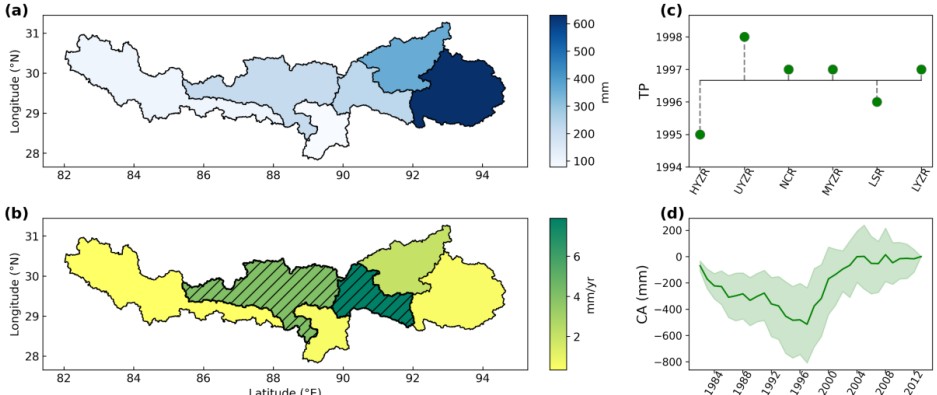

**Figure 2.** Long-term water yield changes over the six sub-basins, covering the entire UBR basin. **(a)** The mean annual values by averaging water yield from 1982 to 2013. **(b)** The temporal variation trends detected by the Mann-Kendall Sen's slope. The black hatching represents statistically significant ($p < 0.05$) trends. **(c)** The turning points (TP) as detected by the Pettitt method. **(d)** The cumulative water yield anomaly (CA) curve. The solid green line represents the ensemble expectation of the cumulative water yield anomaly curves for the entire UBR basin (green shading).

We used the Pettitt method to identify the TPs in the WY over the entire UBR basin. The TPs mainly occurred during the late 1990s; however, the abrupt change detected in some sub-basins was not statistically significant (Fig. 2c and Table. S2). Similarly, the cumulative anomaly curve (Fig. 2d) showed that WY decreased prior to the late 1990s and then increased over the entire UBR basin, which further complemented the results obtained from the Pettitt method. Our results agree with lake area changes in the Tibetan Plateau (Zhang et al., 2017) and climate shifts in the UBR basin (Li et al., 2019b).

**3.2 Regime shifts in historical water yield**

Based on the TPs, we divided the study period from 1982 to 2013 into before and after TP periods, and analyzed the magnitude and direction of the WY changes over the entire UBR basin. Figure 3 shows that the WY increased from 9.5 to 130.9 mm, with high spatial variability. The slight increase observed in the HYZR and LYZR basins accounted for less than 10% of the mean annual water yield before the TP. Nevertheless, a substantial increase in WY of 61.6% and 80.5% was found in the UYZR and MYZR basins, respectively. In addition, higher standard deviations were detected for WY after TP, suggesting more dramatic variability in the entire UBR basin in later years.

For the direction of the WY changes, we found that the change in WY was positive before the TP but became negative afterward in most sub-basins. A significant decreasing trend was detected after the TP in the UYZR, NCR, and LSR basins. In contrast, although the WY in the MYZR basin increased during two periods, the rate of increase had slowed, as the positive trend after the TP (3.64 mm yr$^{-1}$, $p > 0.05$) was less than that before the TP (8.95 mm yr$^{-1}$, $p < 0.05$). Overall, we found that regime shifts in the WY occurred in the late 1990s over the entire UBR basin; the magnitude of the WY changes generally increased, while the direction of the changes reversed or slowed.



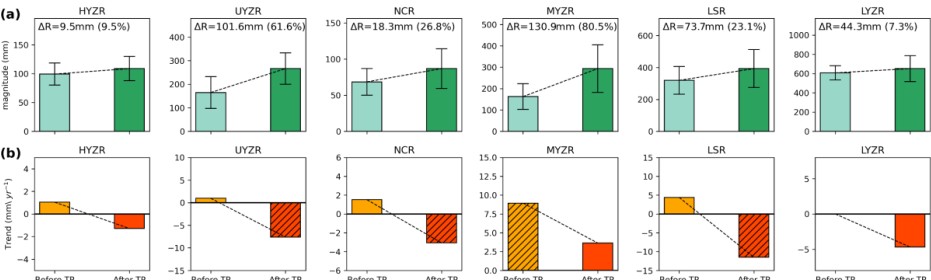

**Figure 3.** Water yield regime shifts over the entire UBR basin. **(a)** Magnitude of the water yield changes. The error bars represent the standard deviation of the water yield before (light green) and after turning point (TP) (green). **(b)** Direction of the water yield changes. The black hatching represents a statistically significant ($p < 0.05$) trend.

**3.3 Attribution analysis on magnitude increases in water yield**

As shown in Fig. 4, we quantified the contributions from climate (eP), vegetation (LAI), and the cryosphere on the WY magnitude increases over the entire UBR basin. We found that the changes in the cryosphere contributed to over half of the magnitude increases in the HYZR, UYZR, NCR, and MYZR basins. However, climate played a more important role in the magnitude increase in the LSR and LYZR basins, with relative contributions of 55.4% and 46.0%, respectively. In contrast to the dominant roles of the climate and cryosphere, vegetation had a consistently positive contribution to the magnitude increases in WY over the entire UBR basin, although the relative contributions of 5.6% in the HYZR basin and 19.9% in the LYZR basin were much less than those from the changes in the climate and cryosphere.

The climate and cryosphere – two important factors influencing the magnitude change in WY – together contributed over 80% to the magnitude increases over the entire UBR basin; however, they played both additive or offsetting roles (Fig. 4), resulting in slight or substantial WY increases (Fig. 3). For example, although the cryosphere change resulted in increases of 28.3 mm and 30.3 mm in the HYZR and NCR basins, the negative contributions from climate offset a considerable part of these increases resulting in the slight increase after the TP in these regions. Additionally, the positive contribution from climate offset the negative contribution from the cryosphere in the LSR and LYZR basins, which resulted in a similar slight increase in WY. However, the additive effects from the climate and cryosphere change lead to substantial increases in WY from 162.6 mm to 293.5 mm in the MYZR basin and from 164.9 mm or 266.5 mm in the UYZR basin.



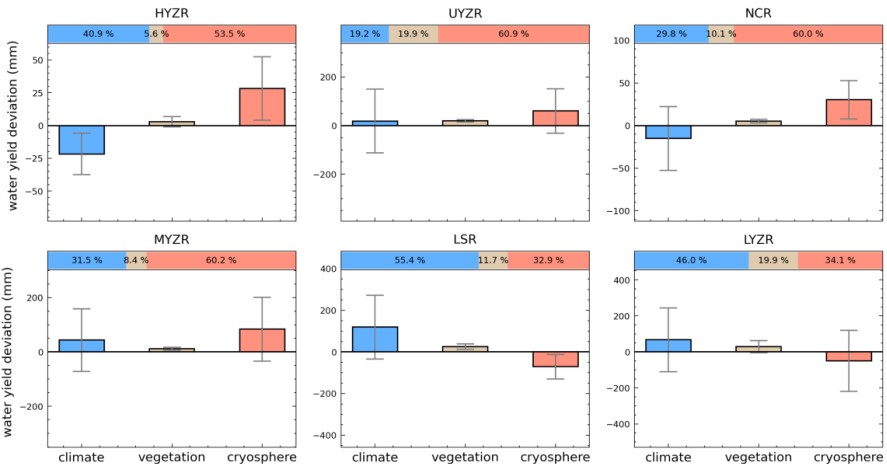

**Figure 4.** Attribution analysis of the magnitude increase in the water yield due to climate (blue bar), vegetation (tan bar), and the cryosphere (red bar), and their relative contributions (the bar on the top) in each basin. The error bars represent the standard deviation of the water yield changes caused by the various drivers (see Fig. S2).

**3.4 Attribution analysis on direction shifts in water yield**

In this study, Pearson's correlation coefficient was applied to determine the role of the climate, vegetation, and cryosphere in the reversed or slowed WY trend after the TPs, as shown in Fig. 3b. The climate played a dominant positive role in influencing the direction of the WY changes after the TP in most sub-basins (Fig. 5), which was supported by correlations ranging from 0.41 (LYZR basin) to 0.93 (LSR basin). However, the changes in WY induced by the cryosphere instead determined the decreasing trend in WY over the HYZR basin ($r = 0.76$, $p < 0.05$). Compared to the significantly positive role of climate, however, cryosphere-induced changes in WY in the UYZR, NCR, and LSR basins exhibited a negative correlation with the decreased WY after the TP. This suggests that meltwater from the cryosphere alleviated the loss of water resources in these regions. In addition, this effect was also detected in the MYZR basin, and together with that of climate, contributed to the increasing trend in the WY in this sub-basin. Despite the weak contribution from vegetation compared to that of the other two drivers (Fig. 4), its positive role in WY decline after the TPs was more apparent in the drier sub-basins (such as HYZR, UYZR, and NCR), whereas the correlation was negative in the humid LYZR basin.

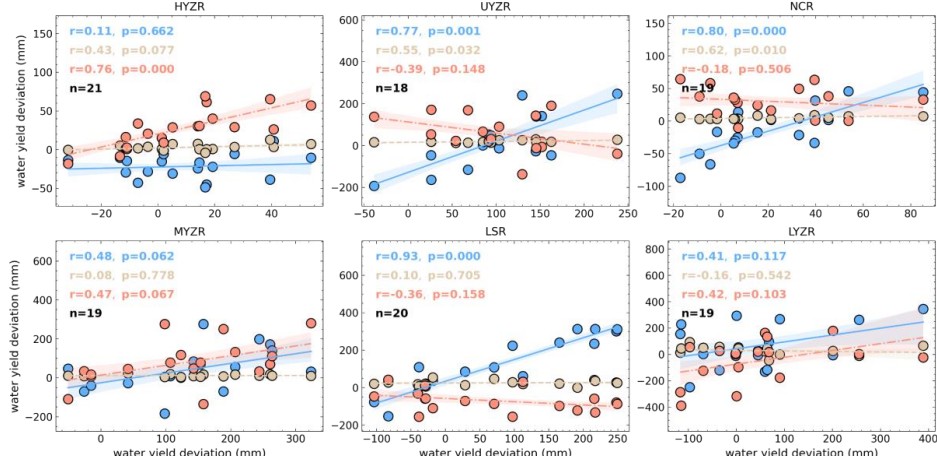

**Figure 5.** The relationship between the time-series of the total water yield deviation ($\Delta WY(t)$, x-axis) and its components (y-axis) induced by climate ($\Delta WY_c(t)$, blue point), vegetation ($\Delta WY_v(t)$, tan point), and cryosphere ($\Delta WY_s(t)$, red point), respectively. The shading area indicates the 95% confidence interval of the fitting. $n$ indicates the number of years after the TP, which was determined by the Pettitt method (See Fig. 2c and Table S2).

## 4 Discussion

The changes in water yield can primarily be attributed to climate change and the cryosphere; nevertheless, they are affected by a complex variety of factors (Harris et al., 2018; Liu et al., 2020; Peng et al. 2017), such as vegetation, snow cover, permafrost, hydrology, and soil properties. Accurate monitoring of cryospheric processes is essential for understanding the changing composite interactions in alpine regions and predicting regional responses to climate warming (Yao et al., 2019). Although some in situ observations have included more physical variables, such as soil moisture and temperature monitoring networks in Naqu and Pali (Chen et al. 2017) and observations of snow and glacial melt runoff in glacier-fed basins (Zhang et al. 2016), there remain large unassessed areas in the UBR basin. The harsh climate and environmental conditions in these regions remain quite challenging to accurate cryosphere-hydrological modeling. In this study, with the support of the Hydrology and Water Resources Survey Bureau of the Tibet Autonomous Region, we collected long-term runoff-gauge data throughout the UBR, examined historical water yield changes, and provided a useful alternative statistical method to physical modeling approaches that can be applied to large-scale alpine river basins to quickly partition the effects of climatic and cryospheric changes on the hydrological regime. Nevertheless, further numerical modeling tools with coupled cryospheric and hydrospheric processes and comprehensive observational data (e.g., Wang et al. 2017) should be developed to better physically and comprehensively understand the mechanisms of the runoff variations in the UBR basin.

Previous studies have demonstrated an increasing trend in WY over the LSR (Lin et al., 2020), LYZR (Zhang et al., 2011), and UBR basins (Li et al., 2021). In this study, we provided further evidence of the long-term trends in WY changes in the above regions, and, furthermore, conducted trend analysis for other regions that have received less attention in the existing literature. Our results comprehensively indicated a general increase in WY (Fig. 2a) over the entire UBR basin. Furthermore, we extended the duration of the





runoff observations to 2013 and found that regime shifts in WY occurred during the late 1990s over the
entire UBR basin. Moreover, the magnitude of WY increased (Fig. 3a), but the direction of WY reversed
or slowed after the TPs (Fig. 3b). To the best of our knowledge, these regime shifts in the WY have not
been reported in previous studies.

Our results indicated that the climate and cryosphere were important factors for magnitude increases

in WY throughout the UBR basin, but their relative contribution varies across regions. Climate explained a
greater increase in WY in downstream regions, while cryospheric changes were more important in
upstream regions (Fig. 4); this matches the relative importance of meltwater from the cryosphere to
streamflow (Fig. S4). According to Biemans et al. (2019), meltwater from the cryosphere is the most
important water source in the upper regions of the Indo-Gangetic Plain, supplying over 40% of the total
WY upstream but less than 30% downstream. The effect of vegetation on changes on WY was much less
than that of the climate and cryosphere (Fig. 4 and Fig. S2). Additionally, offsetting or additive effects
from climate and cryosphere changes were detected in this study (Fig. 4), which led to either slight or
substantial increase in WY in each region of the UBR basin (Fig. 3a). The additive effect is beneficial for
mitigating drought, but it could exacerbate the flood risks due to increased precipitation and accelerated
melting of the cryosphere in the future (Immerzeel et al., 2013). More importantly, the combined effects
often hinder the roles of each driver in hydrological changes, which should be considered when designing
water management strategies and ecological restoration engineering (Wei et al., 2018; Zhang and Wei,

2021).

Although climate and cryosphere together contributed to the magnitude increases in WY throughout

the UBR basin, climate remained the most important factor controlling the declining WY in most regions
(Fig. 5). Simultaneously, significant cryosphere changes due to global warming influenced the direction of
the WY changes, which is supported by glacier retractation (Yao et al., 2010) and several modeling studies
(Lutz et al., 2014; Zhang et al., 2020; Wang Y et al., 2021). Similarly, our study indicated that meltwater
from cryospheric changes has the potential to alleviate reduced water resources in most regions (Fig. 3b).
However, in the HYZR basin, the decline in cryosphere-induced WY became a more important driver of
the decreasing WY trend after the TP, which was inferred from the strong positive correlation ($r = 0.76$, $p$
$< 0.05$, Fig. 5). The meltwater from snow and glaciers in the cryosphere accounted for over 60% of the
streamflow in the HYZR basin (Biemans et al., 2019) and was critical for regional ecology; however, our
statistical results suggested a decreasing supply from the cryosphere after the TP in the HYZR basin, which
could be important for ecological restoration in river sources and emphasizes more explicit physical-based
cryosphere–hydrology modeling.

Effective precipitation, an integrated climatic index that was generated by subtracting the actual

evapotranspiration from the precipitation, was used in the DMC method. As shown in Fig. S3, the mean
annual WY of all six sub-basins showed a consistently linear relationship with the corresponding mean
annual precipitation, further proving the dominant role of precipitation in the spatial and temporal
characteristics of the WY throughout the UBR basin. In addition, Wei et al. (2010, 2018) and Zheng et al.
(2009) conducted attribution analyses of the streamflow caused by climate and land surface changes in
large-scale river basins with mountains and diverse vegetation; they indicated that streamflow variation and
climate variability show a linear relationship, which provides solid evidence for the assumption of a linear



relationship between the WY variation and effective precipitation in the present study. Furthermore, the
results prove that the effects of climate variability could be successfully separated to present a clearer
picture of the cumulative and annual effects of the cryosphere and vegetation changes on the WY in the
UBR basin.
**5 Conclusions**
In this study, regime shifts in WY were detected during the late 1990s over the UBR basin. The magnitude
of the WY generally increased, but its direction reversed or slowed. We used the DMC method to assess
the effects of the climate, vegetation, and cryosphere on the WY and found that the changes in the climate
and cryosphere had either an offsetting or additive effect, which caused either a slight or substantial
increase in the WY, whereas the role of vegetation was much smaller. Furthermore, the declining or
slowing WY after the TPs was mainly driven by climate in most regions, and notably, meltwater from the
cryosphere had the potential to alleviate reduced water resources. These findings suggest that the combined
effects of the climate and cryosphere should be considered in the sustainable development of water
resources and ecosystems, especially the co-benefits in upstream and downstream regions.
*Data availability.* The datasets generated for this study are available on request to the corresponding author.
*Author contributions.* HL: conceptualisation, data curation, formal analysis, methodology, writing –
original draft, writing – review and editing. LL: conceptualisation, formal analysis, methodology, writing –
review and editing, funding acquisition. BYS: data curation, methodology. LW: supervision, writing –
review and editing. AK: validation, writing – review and editing. FZ&DFL: software, validation. XXW:
visualization. WFL&XPL: Writing – review & editing. ZXX: supervision, resources.
*Competing interests.* The contact author has declared that neither they nor their co-authors have any
competing interests.
*Acknowledgments.* This work was jointly supported by the National Natural Science Foundation of China
(Grant No. 51961145104, 52079138, and 91647202), the 2115 Talent Development Program of China
Agricultural University (00109019), and the China Scholarship Council (Grant No. 202006350051). Hao
Li thanks the China Scholarship Council (CSC) for providing financial support to pursue his PhD in
Belgium.



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
