# Peer review of "Significant regime shifts of historical water yield in the Upper Brahmaputra River basin"

_Hydrology and Earth System Sciences, 2022_

## Author Comment (AC1)

Response to Reviewer Comments: **Climate and Cryosphere Cause Water Yield Regime Shifts in the Upper Brahmaputra River basin**

**Reviewer 2**

**Comment 2.1:** The manuscript from Li et al. offers an analysis of the different drivers of hydrological regime shifts in the upper Brahmaputra, highlighting changing climate and glacier loss as major determinants. While the work is generally well written and shaped, there are some issues/suggestions to be considered before publication:

**Reply:** We thank the reviewer for their constructive comments and specifically for the suggestions on using the idea of "peak water" as a conceptual framework for understanding cryospheric contributions to river flow. **We will clarify some definitions, and give related evidence with "peak water" in the Results and Discussion section.** The reply to all comments is shown in the following.

**General Comments**

**Comment 2.2:** A conceptual framework would greatly help you to shape the storyline. Actually, there are several studies showing regime shifts associated with glacier loss (see Huss & Hoch, 2018; concept of "peak water"). I think that your work provides further evidence in that direction, showing the magnitude of hydrological glacier influence over spatial gradients, and offering an important analysis of the turning points in regime shifts.

**Reply:** We thank the reviewer for suggestions. We have now framed our analysis and discussion within the conceptual framework of "peak water" as suggested you. For example,

Revisions in Results:

**Line 193–201:** In contrast to positive contributions of climate, we find that WY caused by cryosphere exhibits a negative association with reduced total WY in recent years in the UYZR (r = -0.39, p > 0.05) and LSR (r = -0.36, p > 0.05) basins. The negative but weak relationship indicates that melt waters from cryospheric loss may compensate for low flow, and even mitigate water shortage risks, as suggested by Bibi et al. (2018) and Gleick and Palaniappan (2010). Also, the compensating effect from cryosphere is much stronger in the MYZR (r = 0.47, p > 0.05), and together with climate contributions, contributes to the increasing WY trend (Figure 4). Different from other regions, however, the HYZR basin shows a significantly positive relationship between cryospheric contributions and total WY (r = 0.76, p < 0.05), indicating that cryosphere instead of climate leads to the downward trend in headwaters. This signifies that in this region, cryospheric contributions have already passed a maximum supplying to river flow, due to decreased glaciers and snow under continuous warming, which is in agreement with Huss and Hock (2018).

**Line 202–206:** We further analyze the relationship of cryospheric contributions to total WY ($RC_s$) with temperature (Figure S6). In the HYZR basin, WY resulting from cryosphere continues to increase with temperature until a maximum is reached, beyond which cryospheric contribution to total WY begins to decrease. In addition, the compensating effect of melt waters also can be seen clearly in the UYZR, MYZR and LSR basins; WY caused by cryospheric loss keeps a positive relationship with the increase of temperature, further supporting the higher correlation in these basins (Figure 6).

Revisions in Discussion:

**Line 225–234:** However, climate, especially precipitation, remains the most important factor controlling the declining WY trend after the TP in most regions (Figure 6 and S5), and may lead to occurrence of turning points (Figure 3c+d), which is in agreement with previous studies (Li et al., 2021; Wang et al., 2021). This suggests the importance of precipitation and its projections on future hydrological process in mountainous watersheds (Lutz et al., 2014). Also, cryospheric contribution to mountainous hydrology is important – melt waters from glaciers and snow melting can alleviate water resources deficits, mainly caused by decreased precipitation in recent years (Figure 6). This finding is also supported by observed glacier runoff data (Yao et al., 2010) and several modeling studies (Lutz et al., 2014; Zhang et al., 2020; Wang et al., 2021). However, after glacier runoff reaches a maximum, defined as 'peak water' (Gleick and Palaniappan, 2010), cryospheric mass loss cannot sustain the rising meltwaters with atmospheric warming (e.g. the HYZR basin in the study). The decreased glacier areas and associated hydrological changes will substantially affect water resources management.

[Figure]

Figure S6. The relationship between cryospheric contributions to water yield deviations ($RC_s(t)$, see Methods in main text) and annual mean 2m maximum air temperature ($T_{max}$) using the polynomial fitting. The colorbar indicates the years after a TP in individual basins. R square here used to evaluate the fitting goodness is labelled in each panel.

**Comment 2.3:** I found the discussion a little bit lacking. As glaciers ad the climate are the major hydrological drivers, what will it happen when glaciers disappear and the climate has shifted? Also, you should stress that the "climate shifts" are changes in precipitation patterns in your work

**Reply:** We thank the reviewer for pointing this out.

We have added additional analysis and discussion of results within the framework of "peak water" as suggested in the previous comment.

Revisions in Results:

**Line 196–201:** Also, the compensating effect from cryosphere is much stronger in the MYZR (r = 0.47, p > 0.05), and together with climate contributions, contributes to the increasing WY trend (Figure 4). Different from other regions, however, the HYZR basin shows a significantly positive relationship between cryospheric contributions and total WY (r = 0.76, p < 0.05), indicating that cryosphere instead of climate leads to the downward trend in headwaters. This signifies that in this region, cryospheric contributions have already passed a maximum supplying to river flow, due to decreased glaciers and snow under continuous warming, which is in agreement with Huss and Hock (2018).

**Line 202–206:** We further analyze the relationship of cryospheric contributions to total WY ($RC_s$) with temperature (Figure S6). In the HYZR basin, WY resulting from cryosphere continues to increase with temperature until a maximum is reached, beyond which cryospheric contribution to total WY begins to decrease. In addition, the compensating effect of melt waters also can be seen clearly in the UYZR, MYZR and LSR basins; WY caused by cryospheric loss keeps a positive relationship with the increase of temperature, further supporting the higher correlation in these basins (Figure 6).

Revisions in Discussion:

**Line 228–234:** Also, cryospheric contribution to mountainous hydrology is important – melt waters from glaciers and snow melting can alleviate water resources deficits, mainly caused by decreased precipitation in recent years (Figure 6). This finding is also supported by observed glacier runoff data (Yao et al., 2010) and several modeling studies (Lutz et al., 2014; Zhang et al., 2020; Wang et al., 2021). However, after glacier runoff reaches a maximum, defined as 'peak water' (Gleick and Palaniappan, 2010), cryospheric mass loss cannot sustain the rising meltwaters with atmospheric warming (e.g. the HYZR basin in the study). The decreased glacier areas and associated hydrological changes will substantially affect water resources management.

Indeed, climate used in the DMC is represented by effective precipitation (eP, P-AET), which is mainly determined by precipitation. We have added related descriptions to highlight the importance of precipitation.

Revisions in Results:

**Line 185–190:** Results in Figure 6 show that, although the correlation varies greatly across basins ranging from 0.11 to 0.93 after the TP, climate typically is positively associated with total WY, in which the correlation is significant in half of basins (p < 0.05), again revealing the major role of climate in the hydrological trends in the entire UBR basin. Further analysis shows that,

precipitation is much more important, because it exhibits the stronger reverse in trend compared with that in actual evaporation (Figure S5), which is also similar with direction changes in WY (Figure 4b).

Revisions in Discussion:

**Line 215–218:** However, climate, especially precipitation, remains the most important factor controlling the declining WY trend after the TP in most regions (Figure 6 and S5), and may lead to occurrence of turning points (Figure 3c+d), which is in agreement with previous studies (Li et al., 2021; Wang et al., 2021). This suggests the importance of precipitation and its projections on future hydrological process in mountainous watersheds (Lutz et al., 2014).

[Figure]

Figure S5. Direction of precipitation (a) and actual evaporation (b) changes. The black hatching represents the statistically significant trend (p < 0.05). The color of boxes represents the period before (light color) and after (dark color) the turning point (TP).

**Comment 2.4:** Why did you choose this particular type of analysis to estimate drivers of regime changes? This is not sufficiently explained in the text

**Reply:** Thank you for your suggestions. We will provide reasons for the use of DMC method in the Introduction and Methods sections.

Revisions in Introduction:

**Line 47–51:** Lastly, the present inadequate understanding of hydrological responses to complex interactions among climate, vegetation, and cryosphere limits the application of hydrological models in those glacier-fed watersheds (Pellicciotti et al., 2012). While, long-term observed runoff data and recent high-resolution precipitation records may give a pathway for using statistical methods to estimate runoff responses to warming in mountainous regions.

Revisions in Methods:

**Line 101–108:** The DMC used here is a plot of the cumulative data of one variable versus the cumulative data of another related variable in a concurrent period. It has previously been used to assess the individual effect of climate (Gao et al., 2011), forest disturbance (Wei and Zhang,

2010), wildfire (Hallema et al., 2018), and cryosphere (Brahney et al., 2017) on water resources. For the large and pristine UBR and other mountainous basins, climate, vegetation, and cryosphere play important roles in hydrology, and these three parts must be together considered to accurately estimate hydrological responses to warming. It is considerably hard to directly calculate the supply of melt waters to WY due to the lack of glacier monitoring, while long-term runoff observations and high-resolution climate and vegetation data make it possible to use the DMC technique, a data-driven statistical method, to estimate cryospheric contribution to WY.

**Line 109–115:** The selection of climate and vegetation indices used in the DMC technique is an important issue. Previous studies haves shown that effective precipitation (eP, P-AET) can reflect more information of climate on WY compared with individual P or AET, and be regarded as a reliable proxy to climate (Wei and Zhang, 2010; Zhang et al., 2019). LAI quantifies the amount of leaf area in an ecosystem and becomes an important variable reflecting vegetation structures and biophysical processes (Fang et al., 2019; Forzieri et al., 2020), and Li et al. (2021) has used LAI to investigate vegetation effects on seasonal hydrology in the UBR basin. Hence, we consider eP and LAI as the indices of climate and vegetation respectively, and use their time series as the inputs in the DMC model.

**Line 116–118:** To obtain cryospheric contribution to WY, we firstly build two types of DMC plots (see Figure S2) to assess the contribution of climate (eP) and vegetation (LAI), and then subtract the sum of estimated contributions from total WY deviations (results are shown in Figure 2). The calculation process of the DMC is shown as follows:

**Comment 2.5:** It seems that the drivers of regime shifts depend on the considered part of the catchment. In general, the influence from glaciers is higher in the upper part and that from precipitation is higher at downstream locations. I think that translating this information into "spatial gradients/turning points" would greatly improve the quality of your work. Is there any relationship between the glacier cover in the catchment and the role of glaciers in driving the magnitude and direction of regime shifts associated with glacier loss or precipitation changes?

**Reply:** The turning point is both controlled by climate and cryospheric loss, and thus it may be not possible to directly build relationships between turning points and cryospheric contributions. In addition, the limited data (we only access snow and glacier area in 2015) may hinder the analysis between glacier areas and cryospheric contributions from the DMC method.

But, Figure 1 in Huss and Hock (2018) recommended by the reviewer shows the responses of cryospheric contributions to river flow under global warming. Based on this, we try to link cryospheric contributions with temperature changes, and find a nonlinear relationship between them (see Figure S6). Related results and discussions are described in the main text.

Revisions in Results:

**Line 202–206:** We further analyze the relationship of cryospheric contributions to total WY ($RC_s$) with temperature (Figure S6). In the HYZR basin, WY resulting from cryosphere continues to increase with temperature until a maximum is reached, beyond which cryospheric contribution to total WY begins to decrease. In addition, the compensating effect of melt waters also can be seen clearly in the UYZR, MYZR and LSR basins; WY caused by cryospheric loss keeps a positive relationship with the increase of temperature, further supporting the higher correlation in these basins (Figure 6).

17

[Figure]

Figure S6 The relationship between cryospheric contributions to water yield deviations ($\Delta WY_s(t)$) and annual mean 2m maximum air temperature ($T_{max}$) using the polynomial fitting. R square used to evaluate the fitting goodness is labelled in each panel.

Revisions in Discussion:

**Line 228–234:** Also, cryospheric contribution to mountainous hydrology is important – melt waters from glaciers and snow melting can alleviate water resources deficits, mainly caused by decreased precipitation in recent years (Figure 6). This finding is also supported by observed glacier runoff data (Yao et al., 2010) and several modeling studies (Lutz et al., 2014; Zhang et al., 2020; Wang et al., 2021). However, after glacier runoff reaches a maximum, defined as 'peak water' (Gleick and Palaniappan, 2010), cryospheric mass loss cannot sustain the rising meltwaters with atmospheric warming (e.g. the HYZR basin in the study). The decreased glacier areas and associated hydrological changes will substantially affect water resources management.

**Abstract**

**Comment 2.6:** I suggest you to remove useless adverbs such as "however, nevertheless, etc.". Try

[Figure]

Figure S6 The relationship between cryospheric contributions to water yield deviations ($\Delta WY_s(t)$) and annual mean 2m maximum air temperature ($T_{max}$) using the polynomial fitting. R square used to evaluate the fitting goodness is labelled in each panel.

to shorten the abstract a little bit, e.g., discarding not essential sentences or summarizing some concepts

**Reply:** Thanks. We have rewritten the abstract (see below).

**Comment 2.7:** Line 15. Change "melt" with "loss". Cryospheric changes can increase the amount of available water, e.g. in rock glaciers

**Reply:** Thanks. We have rewritten the abstract (see below).

**Comment 2.8:** Line 18. Is it "stream head" correct word? I would delete the part ", as represented...downstream" in this sentence, useless for the abstract in my opinion.

**Reply:** Thanks. We have deleted the useless sentence for the abstract to ensure it convey main information to readers (see below).

**Comment 2.9:** Line 19. Delete "we found that"

**Reply:** Thanks. We have rewritten the abstract (see below).

**Comment 2.10:** Line 21. Delete "furthermore"

**Reply:** Thanks. We have rewritten the abstract (see below).

**Comment 2.11:** Line 23. Delete "however"

**Reply:** Thanks. We will delete it (see below).

**Comment 2.12:** Line 25. Delete "nevertheless, we found that"

**Reply:** Thanks. We have rewritten the abstract to strengthen its coherence (see below).

**Comment 2.13:** Line 28. What do you mean with "ecological restoration"? I would remove this word as "water management" is enough in this context. Either, you can use the word "water governance", which involves the management of water as well as the related ecosystems and resources

**Reply:** Thanks for your advice. We deleted "ecological restoration" throughout the manuscript (see below).

The revised abstract is shown:

**Line 1–15:** Although evidence of hydrological responses to climate is abundant, the reliable assessments of water yield (WY) in mountainous watersheds remain unclear due to intensified cryospheric changes. Here we examine long-term WY changes during 1982–2013 in the Upper Brahmaputra River (UBR) basin on the basis of annual runoff observations. Results show that hydrological regime shifts have occurred in the late 1990s; magnitude increases in WY range from ∼10% to ∼80%, while the directions of WY changes reverse from upward to downward after the late 1990s. We then use the double mass curve (DMC) technique to assess the effects of climate, vegetation, and cryosphere on WY regime shifts. Results show that climate and cryosphere together contribute to over 80% of magnitude increases of WY in the entire UBR basin, in which the role of vegetation is nearly negligible. The combined effects, however, are either offsetting or additive, leading to slight or substantial magnitude increases, respectively. Climate change, particularly precipitation decline leads to the downward WY trend in recent years, while melt waters from cryospheric changes may alleviate water shortage in some watersheds. In headwaters, however, cryospheric contributions to WY have declined due to reduced glaciers and snow under warming. Therefore, the combined effects of climate and cryosphere on WY should be considered in water resources management in mountainous watersheds, particularly involving co-benefits for upstream and downstream regions.

**Introduction**

**Comment 2.14:** Line 36. What do you mean with glacial snowmelt? Cryospheric drivers are snow and glaciers providing water across the melting process, i.e., glacier ice melt and snowmelt. Or do you mean the snowmelt occurring on the glacier surface? Please consider here the paper from Huss et al., 2017, which also includes the permafrost ice as a key component of the mountain cryosphere

**Reply:** Thank you pointing it out. We use **"glacier and snow melting"** throughout the revised manuscript, which will be calculated using the DMC method.

**Comment 2.15:** Line 36. It is actually unclear to a layman what the Third Pole is. Please clearly and concisely define it the first time you name it

**Reply:** Thanks. "Third pole" has the less role in the manuscript and also cause the confusion to the abbreviation of "turning point", so we plan to delete it.

**Comment 2.16:** Line 52. "direction of change..." of what?

**Reply:** We will use the following expression to replace it:

**Line 30–31:** For example, Fan and He (2015) highlighted the important role of precipitation in WY increases in the Salween and Mekong River basins.

**Comment 2.17:** Line 54. "glacial snow". See same comment of line 36.

**Reply:** Thanks. We will use **"glacier and snow melting"** throughout the manuscript.

**Comment 2.18:** Line 84-85. "a reference...modelling". You already provided this sentence 8 lines earlier. Please avoid repetition.

**Reply:** Thanks. We will delete the repeated expressions.

**Results**

**Comment 2.19:** Figure 3. I think the use of boxplots would greatly help interpretation.

**Reply:** We will use the boxplots to show the data distribution.

[Figure]

Figure 4. Water yield regime shifts over the entire UBR basin. (a) Magnitude of water yield changes. The colors of boxes represent before (light green) and after the turning point (TP) (green). Black "x" signals show the mean of water yield in each boxplot. (b) Direction of water yield changes. The black hatchings represent the statistically significant trend ($p < 0.05$).

**Comment 2.20:** Figure 5. I suggest you to provide the text and fitting lines for significant relationships only, to avoid figure overwhelming and help interpretation

**Reply:** Thanks for your suggestions. We have created the figure to avoid overwhelming as following.

[Figure]

Figure 6. The relationship between time series of total water yield deviation ($\Delta WY_t(t)$, x-axis) and its components (y-axis) induced by climate ($\Delta WY_c(t)$, blue point), vegetation ($\Delta WY_v(t)$, tan point), and cryosphere ($\Delta WY_s(t)$, red point), respectively. The fitting line and its 95% confidence interval are shown only when p value $< 0.05$. $n$ indicates the number of years after the TP, which is determined by the Pettitt method (See Table 1 and Figure 3c).

**Discussion**

**Comment 2.21:** I think that an important work to be considered, that may help contextualising your storyline, would be Huss & Hock (2018) providing the concept of "peak water". I suggest you to reshape the discussion around this work. Your results clearly show that the upper Brahmaputra has already surpassed the Peak Water and is now in declining phase of hydrological changes associated with glacier loss. This conceptualisation would also help you to better discuss the turning points that different areas experienced during distinct years. . . the presence of these turning points should be better discussed, as it is one of the strengths of the chosen methodology.

**Reply:** Thanks. We have linked the results with "peak water", as revealed in (Huss and Hock, 2018).

Revisions in Results:

**Line 193–201:** In contrast to positive contributions of climate, we find that WY caused by cryosphere exhibits a negative association with reduced total WY in recent years in the UYZR (r = -0.39, p > 0.05) and LSR (r = -0.36, p > 0.05) basins. The negative but weak relationship indicates that melt waters from cryospheric loss may compensate for low flow, and even mitigate water shortage risks, as suggested by Bibi et al. (2018) and Gleick and Palaniappan (2010). Also, the compensating effect from cryosphere is much stronger in the MYZR (r = 0.47, p > 0.05), and together with climate contributions, contributes to the increasing WY trend (Figure 4). Different from other regions, however, the HYZR basin shows a significantly positive relationship between cryospheric contributions and total WY (r = 0.76, p < 0.05), indicating that cryosphere instead of climate leads to the downward trend in headwaters. This signifies that in this region, cryospheric contributions have already passed a maximum supplying to river flow, due to decreased glaciers and snow under continuous warming, which is in agreement with Huss and Hock (2018).

**Line 202–206:** We further analyze the relationship of cryospheric contributions to total WY ($RC_s$) with temperature (Figure S6). In the HYZR basin, WY resulting from cryosphere continues to increase with temperature until a maximum is reached, beyond which cryospheric contribution to total WY begins to decrease. In addition, the compensating effect of melt waters also can be seen clearly in the UYZR, MYZR and LSR basins; WY caused by cryospheric loss keeps a positive relationship with the increase of temperature, further supporting the higher correlation in these basins (Figure 6).

Revisions in Discussion:

**Line 228–234:** Also, cryospheric contribution to mountainous hydrology is important – melt waters from glaciers and snow melting can alleviate water resources deficits, mainly caused by decreased precipitation in recent years (Figure 6). This finding is also supported by observed glacier runoff data (Yao et al., 2010) and several modeling studies (Lutz et al., 2014; Zhang et al., 2020; Wang et al., 2021). However, after glacier runoff reaches a maximum, defined as 'peak water' (Gleick and Palaniappan, 2010), cryospheric mass loss cannot sustain the rising meltwaters with atmospheric warming (e.g. the HYZR basin in the study). The decreased glacier areas and associated hydrological changes will substantially affect water resources management.

**Comment 2.22:** Line 309-311. This is not true! See works from Huss and Hoch (2018), and in general the latest IPCC report on the ocean and the cryosphere, chapter dedicated to mountain environments...

**Reply:** Thank you for pointing it out. In the original manuscript, we want to show that this study provides a more detailed analysis for water yield changes in the UBR basin. We have deleted it to avoid the confusion.

**Comment 2.23:** Line 330. Please change "retractration" with "loss" or "recession". Also the sentence is unclear as written

**Reply:** Thanks. We will use "glacier mass loss".

**Comment 2.24:** Line 338. What do you mean with "ecological restoration"? It is unclear why this would help ecological restoration in particular. It is a general issue of water governance, after all, not just restricted to ecological restoration.

**Reply:** We will use "water resources management" to indicate the implication of this study.

**References**

Bibi, S., Wang, L., Li, X., Zhou, J., Chen, D., and Yao, T.: Climatic and associated cryospheric, biospheric, and hydrological changes on the Tibetan Plateau: A review, International Journal of Climatology, 38, e1–e17, 2018.

Brahney, J., Menounos, B., Wei, X., and Curtis, P. J.: Determining annual cryosphere storage contributions to streamflow using historical hydrometric records, Hydrological Processes, 31, 1590–1601, 2017.

Fan, H. and He, D.: Temperature and precipitation variability and its effects on streamflow in the upstream regions of the Lancang–Mekong and Nu–Salween Rivers, Journal of Hydrometeorology, 16, 2248–2263, 2015.

Fang, H., Baret, F., Plummer, S., and Schaepman-Strub, G.: An overview of global leaf area index (LAI): Methods, products, validation, and applications, Reviews of Geophysics, 57, 739–799, 2019.

Forzieri, G., Miralles, D. G., Ciais, P., Alkama, R., Ryu, Y., Duveiller, G., Zhang, K., Robertson, E., Kautz, M., Martens, B., et al.: Increased control of vegetation on global terrestrial energy fluxes, Nature Climate Change, 10, 356–362, 2020.

Gao, P., Mu, X.-M., Wang, F., and Li, R.: Changes in streamflow and sediment discharge and the response to human activities in the middle reaches of the Yellow River, Hydrology and Earth System Sciences, 15, 1–10, 2011.

Gleick, P. H. and Palaniappan, M.: Peak water limits to freshwater withdrawal and use, Proceedings of the National Academy of Sciences, 107, 11 155–11 162, 2010.

Hallema, D. W., Sun, G., Caldwell, P. V., Norman, S. P., Cohen, E. C., Liu, Y., Bladon, K. D., and McNulty, S. G.: Burned forests impact water supplies, Nature Communications, 9, 1–8, 2018.

Huss, M. and Hock, R.: Global-scale hydrological response to future glacier mass loss, Nature Climate Change, 8, 135–140, 2018.

Li, H., Liu, L., Koppa, A., Shan, B., Liu, X., Li, X., Niu, Q., Cheng, L., and Miralles, D.: Vegetation greening concurs with increases in dry season water yield over the Upper Brahmaputra River basin, Journal of Hydrology, 603, 126–981, 2021.

Lutz, A., Immerzeel, W., Shrestha, A., and Bierkens, M.: Consistent increase in High Asia's runoff due to increasing glacier melt and precipitation, Nature Climate Change, 4, 587–592, 2014.

Pellicciotti, F., Buergi, C., Immerzeel, W. W., Konz, M., and Shrestha, A. B.: Challenges and uncertainties in hydrological modeling of remote Hindu Kush–Karakoram–Himalayan (HKH) basins: suggestions for calibration strategies, Mountain Research and Development, 32, 39–50, 2012.

Wang, L., Yao, T., Chai, C., Cuo, L., Su, F., Zhang, F., Yao, Z., Zhang, Y., Li, X., Qi, J., et al.: TP-River: Monitoring and quantifying total river runoff from the Third Pole, Bulletin of the American Meteorological Society, 102, E948–E965, 2021.

Wei, X. and Zhang, M.: Quantifying streamflow change caused by forest disturbance at a large spatial scale: A single watershed study, Water Resources Research, 46, 2010.

Yao, T., Li, Z., Yang, W., Guo, X., Zhu, L., Kang, S., Wu, Y., and Yu, W.: Glacial distribution and mass balance in the Yarlung Zangbo River and its influence on lakes, Chinese Science Bulletin, 55, 2072–2078, 2010.

Zhang, L., Nan, Z., Wang, W., Ren, D., Zhao, Y., and Wu, X.: Separating climate change and human contributions to variations in streamflow and its components using eight time-trend methods, Hydrological Processes, 33, 383–394, 2019.

Zhang, Y., yu Xu, C., Hao, Z., Zhang, L., Ju, Q., and Lai, X.: Variation of Melt Water and Rainfall Runoff and Their Impacts on Streamflow Changes during Recent Decades in Two Tibetan Plateau Basins, Water, 12, 3112, 2020.

---

## Author Comment (AC2)

Response to Reviewer Comments: **Climate and Cryosphere Cause Water Yield Regime Shifts in the Upper Brahmaputra River basin**

**Reviewer 1**

**Comment 1.1:** Li et al. provide a case study to study the influence of climate and cryosphere on the water yield in the Brahmaputra. To do this they collected a long time series for climatic and precipitation data and analyzed it to find the water yield has changed over the time period studied. They find that there are substantial changes and attribute this mainly to the combined effect of climate and cryosphere. I think this study has value (I especially like the introduction) and should be considered for publication. However, I have several issues that I think should be addressed.

**Reply:** We are grateful for Dr. Florian Ulrich Jehn's thoughtful evaluation and support of our work. We will reply to all comments in detail in the following, highlighting changes in the revised manuscript. **We aim to clarify the reasons for selections of climate and vegetation indices, and double mass curve (DMC) in this study**, which will improve the quality of the revised version.

**General Comments**

**Comment 1.2:** First, the study does not provide enough information about its data. For example, after reading the study I am still unsure what exactly is meant when the study talks about climate being a major factor in its analysis. Is it the mean temperature? Is it some indices? Is it something completely different? Same goes for the term cryosphere, which is used quite loosely.

**Reply:** We thank the reviewer for pointing this out. We now clearly define the variable that we consider as a proxy for the effect of climate on water yield.

Based on water balance ($P = R + AET + \Delta S$), the net effect of climate on regional water yield is expressed in both precipitation ($P$) and actual evaporation ($AET$). Hence, we use effective precipitation ($eP$, $P$ - $AET$) to assess climate contributions to hydrological changes in the double mass curve (DMC) analysis carried out in this study. Therefore, we consider eP as a proxy to climate, supported by Wei and Zhang (2010) and Zhang et al. (2019). In the revised version, we clearly explain the reasons for selecting eP as a proxy for climate effects as follows:

**Line 109–111:** The selection of climate and vegetation indices used in the DMC technique is an important issue. Previous studies haves shown that effective precipitation (eP, P-AET) can reflect more information of climate on WY compared with individual P or AET, and be regarded as a reliable proxy to climate (Wei and Zhang, 2010; Zhang et al., 2019).

Cryospheric contributions to water yield calculated by the DMC method mean that, melt waters released from **glacier and snow melting with warming** contribute to river flow. In DMC, we define cryospheric contributions to water yield as the values of total water yield deviation minus the sums of climate and vegetation contributions. Related revisions are shown as follows:

**Line 116–118:** To obtain cryospheric contribution to WY, we firstly build two types of DMC plots (see Figure S2) to assess the contribution of climate (eP) and vegetation (LAI), and then subtract the sum of estimated contributions from total WY deviations (results are shown in Figure 2). The calculation process of the DMC is shown as follows:

**Comment 1.3:** Second, after reading the methods it is not clear to me how the study is able to differentiate between the influence on climate, cryosphere and vegetation. This section would

profit from a more in depth explanation. In addition, why using this method? Why do you think it is especially good for your kind of study?

**Reply:** We thank the reviewer for this suggestion.

For a large natural mountainous watershed, it still remains not clear about the hydrological responses to climate change and associated environmental changes, e.g. vegetation and cryosphere. That leads to great uncertainties when assessing water yield changes using hydrological models. While, long-term annual runoff observations and high-resolution precipitation records in the UBR basin provide a good opportunity for statistical models to investigate hydrological responses to climate warming.

The Double Mass Curve – a data-driven statistical model – has been widely been applied to estimate water yield responses to environmental changes in the hydrological community. We assume that in the UBR basin, water yield is affected by climate (e.g. precipitation and evaporation), vegetation greening or browning, and cryospheric loss (e.g. glacier and snow melting). Hence, we can estimate cryospheric contributions to water yield by subtracting the sum of contributions from climate and vegetation from total deviations using the DMC method.

We now clearly explain the reasons for using the DMC to separate climate, vegetation and cryosphere contributions to water yield in the manuscript. Related changes in Introduction and Methods section are shown as follows:

Revisions in Introduction:

**Line 47–51:** Lastly, the present inadequate understanding of hydrological responses to complex interactions among climate, vegetation, and cryosphere limits the application of hydrological models in those glacier-fed watersheds (Pellicciotti et al., 2012). While, long-term observed runoff data and recent high-resolution precipitation records may give a pathway for using statistical methods to estimate runoff responses to warming in mountainous regions.

Revisions in methods:

**Line 101–108:** The DMC used here is a plot of the cumulative data of one variable versus the cumulative data of another related variable in a concurrent period. It has previously been used to assess the individual effect of climate (Gao et al., 2011), forest disturbance (Wei and Zhang, 2010), wildfire (Hallema et al., 2018), and cryosphere (Brahney et al., 2017) on water resources. For the large and pristine UBR and other mountainous basins, climate, vegetation, and cryosphere play important roles in hydrology, and these three parts must be together considered to accurately estimate hydrological responses to warming. It is considerably hard to directly calculate the supply of melt waters to WY due to the lack of glacier monitoring, while long-term runoff observations and high-resolution climate and vegetation data make it possible to use the DMC technique, a data-driven statistical method, to estimate cryospheric contribution to WY.

**Comment 1.4:** Third, the study finds a turning point for the behavior of the river. This seems quite important to me, but is never really discussed. Why did this change happen? What consequences will it have?

**Reply:** We thank the reviewer for pointing this out. The turning point is identified by the Pettitt method and used to split the entire period into two parts in which water yield shows substantial

changes in both the magnitude and direction (Section 3.2). In fact, this is a prerequisite step for using the DMC method to assess the relative importance of climate, vegetation and cryosphere in driving hydrological changes between the two periods (see Methods).

As revealed in our study, significant water yield changes in two periods are determined by the turning points, in which climate and cryosphere both contribute to magnitude increases in water yield, but climate, represented by $P - AET$, is more important for the trend changes in water yield. Further, We make related analysis and discussions to stress the dominant role of precipitation in water yield changes in the revised version as follows:

Revisions in Results:

**Line 185–190:** Results in Figure 6 show that, although the correlation varies greatly across basins ranging from 0.11 to 0.93 after the TP, climate typically is positively associated with total WY, in which the correlation is significant in half of basins ($p < 0.05$), again revealing the major role of climate in the hydrological trends in the entire UBR basin. Further analysis shows that, precipitation is much more important, because it exhibits the stronger reverse in trend compared with that in actual evaporation (Figure S5), which is also similar with direction changes in WY (Figure 4b).

[Figure]

Figure S5. Direction of precipitation (a) and actual evaporation (b) changes. The black hatching represents the statistically significant trend ($p < 0.05$). The color of boxes represents the period before (light color) and after (dark color) the turning point (TP).

Revisions in Discussion:

**Line 215–218:** However, climate, especially precipitation, remains the most important factor controlling the declining WY trend after the TP in most regions (Figure 6 and S5), and may lead to occurrence of turning points (Figure 3c+d), which is in agreement with previous studies (Li et al., 2021; Wang et al., 2021). This suggests the importance of precipitation and its projections on future hydrological process in mountainous watersheds (Lutz et al., 2014).

**Specific Comments**

**Comment 1.5:** The study states several times that the increase meltwater has the potential to

alleviate the loss of water availability. I also think this is the case, but it should be made clearer that this will only be a temporary relief until the glaciers have melted.

**Reply:** We agree with the reviewer that meltwater is only a temporary relief. We will improve these statements. As mentioned, there may be a "maximum cryospheric contribution to water yield" ('peak water' Gleick and Palaniappan (2010)). Glacier runoff will increase with warming and compensate for low flow during droughts (see negative correlations with decrease runoff in most basins, Figure 6), while steadily decrease after reaching "peak water" due to the reduced glaciers and snow (see the positive correlation in the HYZR basin, Figure 6a).

In the revised manuscript, we try to link our results with "peak water". For example,

Revisions in Results:

**Line 193–201:** In contrast to positive contributions of climate, we find that WY caused by cryosphere exhibits a negative association with reduced total WY in recent years in the UYZR (r = -0.39, p > 0.05) and LSR (r = -0.36, p > 0.05) basins. The negative but weak relationship indicates that melt waters from cryospheric loss may compensate for low flow, and even mitigate water shortage risks, as suggested by Bibi et al. (2018) and Gleick and Palaniappan (2010). Also, the compensating effect from cryosphere is much stronger in the MYZR (r = 0.47, p > 0.05), and together with climate contributions, contributes to the increasing WY trend (Figure 4). Different from other regions, however, the HYZR basin shows a significantly positive relationship between cryospheric contributions and total WY (r = 0.76, p < 0.05), indicating that cryosphere instead of climate leads to the downward trend in headwaters. This signifies that in this region, cryospheric contributions have already passed a maximum supplying to river flow, due to decreased glaciers and snow under continuous warming, which is in agreement with Huss and Hock (2018).

**Line 202–206:** We further analyze the relationship of cryospheric contributions to total WY ($RC_s$) with temperature (Figure S6). In the HYZR basin, WY resulting from cryosphere continues to increase with temperature until a maximum is reached, beyond which cryospheric contribution to total WY begins to decrease. In addition, the compensating effect of melt waters also can be seen clearly in the UYZR, MYZR and LSR basins; WY caused by cryospheric loss keeps a positive relationship with the increase of temperature, further supporting the higher correlation in these basins (Figure 6).

Revisions in Disucssion:

**Line 228–234:** Also, cryospheric contribution to mountainous hydrology is important – melt waters from glaciers and snow melting can alleviate water resources deficits, mainly caused by decreased precipitation in recent years (Figure 6). This finding is also supported by observed glacier runoff data (Yao et al., 2010) and several modeling studies (Lutz et al., 2014; Zhang et al., 2020; Wang et al., 2021). However, after glacier runoff reaches a maximum, defined as 'peak water' (Gleick and Palaniappan, 2010), cryospheric mass loss cannot sustain the rising meltwaters with atmospheric warming (e.g. the HYZR basin in the study). The decreased glacier areas and associated hydrological changes will substantially affect water resources management.

**Comment 1.6:** What are the specific reasons that vegetation was studied? Are the any reasons to assume that the vegetation has changed significantly in the time period?

**Reply:** Many studies have indicated that vegetation will significantly change water yield based on the statistical or physical models. Also, recent study by Li et al. (2021) revealed that vegetation greening in this region may redistribute water resources through time. Therefore, we use the DMC in the study to estimate vegetation effects on water yield.

**Comment 1.7:** Figure S2 belongs in the paper in my opinion, as it seems like this is your main plot, which all following plots refer to.

**Reply:** We thank the reviewer for this suggestion with which we agree. We have placed it in main text.

**Comment 1.8:** Figure 1: Please change this 3D pie chart to bar char, as those are much easier to read.

**Reply:** Thank you for pointing it out.

**Comment 1.9:** Do the abbreviations that are used to label the subcatchments have any meaning?

**Reply:** Yes. For example, "HYZR" means the headwater watershed in the Yarlung Zangbo River (YZR) basin, and "LSR" means Lasha River basin. We create a table to summarize the information in main text.

**Line 70–71:**

Table 1: Information of six sub-basins divided by the locations of hydrological stations. The column "Abbre." and "Full" mean the abbreviations and full names of sub-basins. The column "River" indicates whether the basin is located in the main or branch river of the Yarlung Zangbo River (YZR). The column "Station", "Lon.", and "Lat." indicate names and geolocations of hydrological stations. The column "Area" means the total area of sub-basins. The column "TP" indicates the turning point using the Pettitt method, in which a significant tuning point is labeled with *.

| Abbrev. | Full name | River | Station | Lon. (°) | Lat. (°) | Area (km²) | TP |
|---------|-----------|-------|---------|----------|----------|-----------|-----|
| HYZR | Headstream | main | Lhatse | 87.57 | 29.12 | 49,739 | 1995 |
| UYZR | Upstream | main | Nugesha | 89.71 | 29.32 | 43,916 | 1998* |
| NCR | Nianchu River | branch | Shigatse | 88.89 | 29.28 | 14,359 | 1997* |
| MYZR | Midstream | main | Yangcun | 91.82 | 29.26 | 20,004 | 1997* |
| LSR | Lhasa River | branch | Lhasa | 91.15 | 29.64 | 25,601 | 1996 |
| LYZR | Downstream | main | Nuxia | 94.65 | 29.46 | 45,017 | 1997 |

**Comment 1.10:** Did you check if you evapotransporiration is roughly correct? You used evapotransporation data from a global model, which might have not been calibrated well to regions such extreme as yours.

**Reply:** We thank the reviewer for pointing this out. As we do not have access to observed actual evaporation data in this region, we used AET from GLEAM. This dataset has been extensively validated across varied vegetation types in China and has shown good performance with in situ observations.

Related revisions are as follows:

**Line 85–87:** Regional actual evaporation (AET) is acquired from Global Land Evaporation Amsterdam Model (GLEAM) (Martens et al., 2017). The evaporation product has been validated in different biome types in China and has shown high correlations with in-situ eddy covariance AET (Yang et al., 2017).

**Comment 1.11:** Why did you choose LAI as a proxy for vegetation and not some other measure?

**Reply:** Thanks for your questions about the selections of vegetation indices. We give the reasons in the revised version.

**Line 111–115:** LAI quantifies the amount of leaf area in an ecosystem and becomes an important variable reflecting vegetation structures and biophysical processes (Fang et al., 2019; Forzieri et al., 2020), and Li et al. (2021) has used LAI to investigate vegetation effects on seasonal hydrology in the UBR basin. Hence, we consider eP and LAI as the indices of climate and vegetation respectively, and use their time series as the inputs in the DMC model.

**Comment 1.12:** Have you considered also checking for the runoff-ratio? This seems like a variable that should give you some additional information.

**Reply:** We thank the reviewer for this suggestion. As effective precipitation is one of the main explanatory variable considered in this study, we decided to use just water yield, or runoff depth as the target variable. This allows us to robustly apply the statistical method (DMC) to quantify hydrological responses to climate warming.

**Comment 1.13:** Please change Fig. 3 and Fig 4. to boxplots or swarmplots (depending on your sample size you calculated your mean and standard deviation from). Having just a bar plot with a standard deviation does not really show how your underlying data looks like.

**Reply:** We agree with the reviewer. We will use boxplots to support our analysis. Note the labelled numbers have been changed.

[Figure]

Figure 4. Water yield regime shifts in the entire UBR basin. (a) Magnitude of water yield changes. Black "x" signals show the mean of water yield in each boxplot. (b) Direction of water yield changes. The black hatching represents the statistically significant trend ($p < 0.05$). The color of boxes represents the period before (light color) and after (dark color) the turning point (TP).

[Figure]

Figure 5. Attribution analysis of magnitude increases in water yield due to climate ($\Delta WY_c$, blue box), vegetation ($\Delta WY_v$, tan box), and cryosphere ($\Delta WY_s$, red box), and their relative contributions (the bar with colors on the top) in each basin. Black "x" signals show the mean of water yield deviations (see Figure 2) in each boxplot.

**Comment 1.14:** Are your p-values corrected? If not, this would mean that likely in Figure 5 there are way fewer significant trends.

**Reply:** Thanks for your comments. The p value here is correct, but I am sorry that the labelled $n$ is wrong due to a minor programming error. In the revised version, we have corrected this error, and will only show the regression lines in Figure 6 when p value is less than 0.05, and use "significant" or "significantly" in main text. Related revisions in the manuscript are as follows:

**Line 185–192:** Pearson's correlation coefficient is applied to determine the role of climate, vegetation, and cryosphere in the reversed or slowed WY trend after the TP, as shown in Figure 4b. Results in Figure 6 show that, although the correlation varies greatly across basins ranging from 0.11 to 0.93 after the TP, climate typically is positively associated with total WY, in which the correlation is significant in half of basins (p < 0.05), again revealing the major role of climate in the hydrological trends in the entire UBR basin. Further analysis shows that, precipitation is much more important, because it exhibits the stronger reverse in trend compared with that in actual evaporation (Figure S5), which is also similar with direction changes in WY (Figure 4b). Additionally, despite the weak contribution of vegetation (Figure 5), its positive role in WY changes is more apparent in the drier

[Figure]

Figure 6. The relationship between time series of total water yield deviation ($\Delta WY_t(t)$, x-axis) and its components (y-axis) induced by climate ($\Delta WY_c(t)$, blue point), vegetation ($\Delta WY_v(t)$, tan point), and cryosphere ($\Delta WY_s(t)$, red point), respectively. The fitting line and its 95% confidence interval are shown only when p value < 0.05. $n$ indicates the number of years after the TP, which is determined by the Pettitt method (See Table 1 and Figure 3c).

basins (such as UYZR, and NCR), while the correlation is negative in the relatively humid LYZR basin.

**Line 193–201:** In contrast to positive contributions of climate, we find that WY caused by cryosphere exhibits a negative association with reduced total WY in recent years in the UYZR (r = -0.39, p > 0.05) and LSR (r = -0.36, p > 0.05) basins. The negative but weak relationship indicates that melt waters from cryospheric loss may compensate for low flow, and even mitigate water shortage risks, as suggested by Bibi et al. (2018) and Gleick and Palaniappan (2010). Also, the compensating effect from cryosphere is much stronger in the MYZR (r = 0.47, p > 0.05), and together with climate contributions, contributes to the increasing WY trend (Figure 4). Different from other regions, however, the HYZR basin shows a significantly positive relationship between cryospheric contributions and total WY (r = 0.76, p < 0.05), indicating that cryosphere instead of climate leads to the downward trend in headwaters. This signifies that in this region, cryospheric contributions have already passed a maximum supplying to river flow, due to decreased glaciers

and snow under continuous warming, which is in agreement with Huss and Hock (2018).

**Comment 1.15:** The text is quite heavy on abbreviations, which makes it harder to read. Please consider just writing the words out instead of abbreviating them.

**Reply:** Thank you for the suggestions. We have deleted some abbreviations in main text, such as "Third Pole (TP)". And we also will provide a table (see Table 1 in main text) to indicate some abbreviations clearly.

**Technical corrections**

**Comment 1.16:** L19-21: I am not able to parse this sentence.

**Reply:** We thank the reviewer for pointing this out. We have rewritten the abstract and convey the main information clearer to readers.

**Line 1–15:** Although evidence of hydrological responses to climate is abundant, the reliable assessments of water yield (WY) in mountainous watersheds remain unclear due to intensified cryospheric changes. Here we examine long-term WY changes during 1982–2013 in the Upper Brahmaputra River (UBR) basin on the basis of annual runoff observations. Results show that hydrological regime shifts have occurred in the late 1990s; magnitude increases in WY range from ∼10% to ∼80%, while the directions of WY changes reverse from upward to downward after the late 1990s. We then use the double mass curve (DMC) technique to assess the effects of climate, vegetation, and cryosphere on WY regime shifts. Results show that climate and cryosphere together contribute to over 80% of magnitude increases of WY in the entire UBR basin, in which the role of vegetation is nearly negligible. The combined effects, however, are either offsetting or additive, leading to slight or substantial magnitude increases, respectively. Climate change, particularly precipitation decline leads to the downward WY trend in recent years, while melt waters from cryospheric changes may alleviate water shortage in some watersheds. In headwaters, however, cryospheric contributions to WY have declined due to reduced glaciers and snow under warming. Therefore, the combined effects of climate and cryosphere on WY should be considered in water resources management in mountainous watersheds, particularly involving co-benefits for upstream and downstream regions.

**Comment 1.17:** L45: would delete this mention of "Third Pole" as this exact phrasing has already been used in the paragraph above it.

**Reply:** Yes. It is not important for this study, and we have deleted it.

**References**

Bibi, S., Wang, L., Li, X., Zhou, J., Chen, D., and Yao, T.: Climatic and associated cryospheric, biospheric, and hydrological changes on the Tibetan Plateau: A review, International Journal of Climatology, 38, e1–e17, 2018.

Brahney, J., Menounos, B., Wei, X., and Curtis, P. J.: Determining annual cryosphere storage contributions to streamflow using historical hydrometric records, Hydrological Processes, 31, 1590–1601, 2017.

Fang, H., Baret, F., Plummer, S., and Schaepman-Strub, G.: An overview of global leaf area index (LAI): Methods, products, validation, and applications, Reviews of Geophysics, 57, 739–799, 2019.

Forzieri, G., Miralles, D. G., Ciais, P., Alkama, R., Ryu, Y., Duveiller, G., Zhang, K., Robertson, E., Kautz, M., Martens, B., et al.: Increased control of vegetation on global terrestrial energy fluxes, Nature Climate Change, 10, 356–362, 2020.

Gao, P., Mu, X.-M., Wang, F., and Li, R.: Changes in streamflow and sediment discharge and the response to human activities in the middle reaches of the Yellow River, Hydrology and Earth System Sciences, 15, 1–10, 2011.

Gleick, P. H. and Palaniappan, M.: Peak water limits to freshwater withdrawal and use, Proceedings of the National Academy of Sciences, 107, 11 155–11 162, 2010.

Hallema, D. W., Sun, G., Caldwell, P. V., Norman, S. P., Cohen, E. C., Liu, Y., Bladon, K. D., and McNulty, S. G.: Burned forests impact water supplies, Nature Communications, 9, 1–8, 2018.

Huss, M. and Hock, R.: Global-scale hydrological response to future glacier mass loss, Nature Climate Change, 8, 135–140, 2018.

Li, H., Liu, L., Koppa, A., Shan, B., Liu, X., Li, X., Niu, Q., Cheng, L., and Miralles, D.: Vegetation greening concurs with increases in dry season water yield over the Upper Brahmaputra River basin, Journal of Hydrology, 603, 126–981, 2021.

Lutz, A., Immerzeel, W., Shrestha, A., and Bierkens, M.: Consistent increase in High Asia's runoff due to increasing glacier melt and precipitation, Nature Climate Change, 4, 587–592, 2014.

Martens, B., Miralles, D. G., Lievens, H., Van Der Schalie, R., De Jeu, R. A., Fernández-Prieto, D., Beck, H. E., Dorigo, W. A., and Verhoest, N. E.: GLEAM v3: Satellite-based land evaporation and root-zone soil moisture, Geoscientific Model Development, 10, 1903–1925, 2017.

Pellicciotti, F., Buergi, C., Immerzeel, W. W., Konz, M., and Shrestha, A. B.: Challenges and uncertainties in hydrological modeling of remote Hindu Kush–Karakoram–Himalayan (HKH) basins: suggestions for calibration strategies, Mountain Research and Development, 32, 39–50, 2012.

Wang, L., Yao, T., Chai, C., Cuo, L., Su, F., Zhang, F., Yao, Z., Zhang, Y., Li, X., Qi, J., et al.: TP-River: Monitoring and quantifying total river runoff from the Third Pole, Bulletin of the American Meteorological Society, 102, E948–E965, 2021.

Wei, X. and Zhang, M.: Quantifying streamflow change caused by forest disturbance at a large spatial scale: A single watershed study, Water Resources Research, 46, 2010.

Yang, X., Yong, B., Ren, L., Zhang, Y., and Long, D.: Multi-scale validation of GLEAM evapotranspiration products over China via ChinaFLUX ET measurements, International Journal of Remote Sensing, 38, 5688–5709, 2017.

Yao, T., Li, Z., Yang, W., Guo, X., Zhu, L., Kang, S., Wu, Y., and Yu, W.: Glacial distribution and mass balance in the Yarlung Zangbo River and its influence on lakes, Chinese Science Bulletin, 55, 2072–2078, 2010.

Zhang, L., Nan, Z., Wang, W., Ren, D., Zhao, Y., and Wu, X.: Separating climate change and human contributions to variations in streamflow and its components using eight time-trend methods, Hydrological Processes, 33, 383–394, 2019.

Zhang, Y., yu Xu, C., Hao, Z., Zhang, L., Ju, Q., and Lai, X.: Variation of Melt Water and Rainfall Runoff and Their Impacts on Streamflow Changes during Recent Decades in Two Tibetan Plateau Basins, Water, 12, 3112, 2020.

---

## Author Response (AR1)

Response to Editors' and Reviewers' Comments: **Climate and Cryosphere Cause Water Yield Regime Shifts in the Upper Brahmaputra River basin**

Dear Dr. Giulia Zuecco,

Thank you for giving us the opportunity to submit a revised draft of our manuscript, now entitled "Climate and Cryosphere Cause Water Yield Regime Shifts in the Upper Brahmaputra River basin" to Journal *Hydrology and Earth System Sciences*. We appreciate the editors and reviewers for devoting their precious time to reviewing our paper and providing valuable suggestions. It was their insightful comments that have led to considerable improvements in the current version.

We have carefully considered your comments and incorporated the required changes, mainly including (1) the schematic diagram explaining the DMC method used here, (2) detailed discussion about the results, especially "peak water", and potential implications on mountainous water resources management, and (3) some limitations and uncertainties in the study. Moreover, the text has been carefully edited to improve the English writing and convey the results of the study more clearly.

Below we provide a point-by-point response to the comments and concerns from editors and reviewers. All modifications in the manuscript have been marked.

Sincerely

Hao Li (on behalf of all coauthors)
PhD candidate, Hao.liwork@ugent.be
Hydro-Climate Extremes Lab, Ghent University
Jun 10, 2022

Li Hao

**Editor**

**Comment 0.1:** Thank you for your responses to the comments provided by the reviewers. Both reviewers had major concerns about the manuscript, and they pointed out some key limitations. Specifically, the first reviewer observed that there is not enough information about the datasets, a clear explanation of the climate and cryosphere terms, and of the methodological approach used to differentiate the influence of climate, cryosphere and vegetation on the water yield. The second reviewer also asked to better address the cryosphere component (is it related only to the glacier ice melt or also to snowmelt and permafrost?), and to improve the discussion of the manuscript. Furthermore, both reviewers suggested improving the discussion about the turning points that were found. I carefully considered both the comments of the reviewers and your responses. Overall, I think that the responses to the first reviewer about the clarification of the two terms, climate and cryosphere, could have been more detailed, as well as the explanation of the methodological approach. Besides the comments and suggestions of the two reviewers (that should be considered in the revision), I have the following recommendations that I would like to see implemented in the revised version of the manuscript.

**Reply:** We are grateful the time and energy from Dr. Giulia Zuecco's and two reviewers in reviewing our manuscript. We have clarified your questions below and incorporated your suggestions into our revised manuscript, which improved significantly the quality of the manuscript. We sincerely hope that the revised version can satisfy the editors and reviewers.

**Comment 0.2:** Please move Table S1 in the main manuscript, and add details about the elevation ranges in the seven catchments, the main land use, and the percentage of glacier ice cover in each catchment (these details can be referred to a specific year, e.g. 2013 or 2015).

**Reply:** Thanks for your suggestion. We have placed it in the main text, and added basic information, e.g. basin area, average elevation, hydrological stations, and the percentage of glacier and snow referred to 2000 land use and cover (Table 1). The information will provide readers with a detailed introduction about our study region.

Table 1: Information of six basins divided by the locations of hydrological stations. The column "TP" indicates the turning point using the Pettitt method, in which a significant tuning point is labeled with *. Glaciers and snow area is acquired from the land use and cover in 2000 (see details in Dataset).

| Abbrevation | Full names | Station | Total area ($km^2$) | Mean elevation (m) | Glaciers and snow percentage (%) | TP |
|---|---|---|---|---|---|---|
| HYZR | Headstream | Lhatse | 49,739 | 5061 | 1.7 | 1995 |
| UYZR | Upstream | Nugesha | 43,916 | 4985 | 0.39 | 1998* |
| NCR | Nianchu River | Shigatse | 14,359 | 4733 | 1.96 | 1997* |
| MYZR | Midstream | Yangcun | 20,004 | 4681 | 1.81 | 1997* |
| LSR | Lhasa River | Lhasa | 25,601 | 4879 | 0.72 | 1996 |
| LYZR | Downstream | Nuxia | 45,017 | 4586 | 2.51 | 1997 |

**Comment 0.3:** Based on Fig. 1c and the catchment areas reported in Table S1, it seems that glaciers and snow cover a very limited area in each catchment. How is it possible that the cryosphere represents such a major component in water yield in most of the catchments presented in Fig.

4, when glaciers and snow cover an area that is less than 5% in each catchment? To understand whether there is a major effect of glacier ice melt or snowmelt on runoff response (and consequently on the water yield), I suggest to the authors to show the time series of runoff in the supplementary material, and a monthly scale analysis of runoff, air temperature and precipitation to support the main findings of the manuscript, and whether you can consider the cryosphere a major component in the hydrology of these catchments.

**Reply:** Thanks for your comments. As you point, glaciers and snow is not the main land use and cover in the UBR basin, while its melt water with global warming is an important supply source for regional water resources (Yao et al., 2019; Wang et al., 2022); meltwater contributes to over 40% of water yield in upper regions but reduces to less than 30% in the downstream (Biemans et al., 2019). In addition, our study may indicate that the basins with minor glaciers and snow coverage can also cause substantial consequences on mountainous hydrological regimes, which is line with Huss and Hock (2018).

On the other hand, to show the importance of melt waters in the UBR basin, we analyze the monthly runoff-precipitation ratio in the Nuxia station (the outlet of the UBR basin in this study, Figure 1). Figure R1 indicates that the ratio varies across seasons substantially; it changes between the range of 0.4 and 0.5 in wet seasons, while it significantly exceeds 1.0 in November and December. That reveals that precipitation is not the decisive factor and instead melt waters may lead to the hydrological process in dry seasons.

Although we are limited by the acquisition of monthly observed data in other hydrological stations, we still can infer that the changes of the seasonal runoff-precipitation ratio may become more significant in other hydrological stations, because of the much more important role of melt waters in hydrology changes in upper basins (Biemans et al., 2019).

[Figure]

Figure R1. Monthly runoff-precipitation ratio observed in the Nuxia hydrological station. The green line represents multi-year average value and the light green lines represent the runoff-precipitation ratio from 1982 to 2013 (32 years).

**Comment 0.4:** Section 2.2: What is the temporal resolution of runoff, climate and vegetation data? In addition, please add in this section information about how the extensions of glacier and snow-covered areas were obtained. Furthermore, I suggest reporting the time series of water yield for all catchments in the main manuscript to show the detected turning points.

**Reply:** Thanks for your comments. In this study, we align the climate and vegetation data with runoff from 1982 to 2013. The description about the temporal resolution of climate and vegetation data used here has been incorporated into the revised version.

**Line 70-73:** Here we collect annual runoff observations from 1982 to 2013 and convert the river flow ($m^3/s$) into runoff depth ($mm$). Also, we acquire high-resolution climate and vegetation data in the same time range, and further aggregate these gridded data into regional annual values by considering area-weighted effects.

Also, we reported the source of land use and cover used in our study in this section.

**Line 93-95:** The land use and cover in 2000 with a spatial resolution of 1x1 km is used to represent the land cover types in the UBR basin. The data is acquired from Resource and Environmental Science Data Center, and is here divided into seven primary land use types, including cultivated land, forestland, grassland, water body, urban land, unused land, and glaciers and snow (Figure S3).

Finally, We have added the annual time-series of water yield for all catchments to show the detected turning points (see Figure S2 in the supplementary), clearly illustrating a reversed or slowed WY changing direction but an increase of WY magnitude in the UBR basin (Figure 4).

[Figure]

Figure S2. The temporal changes of precipitation (P, blue line), actual evaporation (AET, orange line), and water yield (WY, green line) and Leaf Area Index (LAI, grey line) during 1982–2013 in the entire UBR basin. The vertical line indicates the turning point in WY.

**Comment 0.5:** Section 2.3: As stated by the first reviewer, the methodological approach is unclear and difficult to understand. To improve the clarity of this section, I invite the authors to add an

example of the double mass curve for one of the catchments, combined with a scheme that should help the reader to understand how the three components (climate, vegetation and cryosphere) were identified. For an example of what I mean, please see Figs. 2 and 3 in Brahney et al. (2017).

**Reply:** Thanks for your suggestion. We have rewritten the method with the help of the schematic diagram (Figure 2) for the DMC method in the main text (see Details in Methodology).

[Figure]

Figure 2. The schematic diagram showing how to estimate the effects of climate, vegetation, and cryosphere on water yield in the MYZR basin (Details in Methodology).

**Comment 0.6:** Section 2.3.2: In this section it is unclear whether water yield and eP represent the same term (L155) or they were computed in a different way. Based on Section 2.2.1, water yield was obtained using only runoff data, whereas at L149-150, runoff is not mentioned as term of the water balance and for the computation of the water yield. Please rephrase all these sentences to clarify the definition of the two terms and their differences.

**Reply:** We are sorry for this ambiguous description here. Here, water yield comes from observations in hydrological stations, while the effective precipitation (eP, P-AET) is calculated by the high-resolution datasets. In Line 149-150 in the original manuscript, we want to state the reasons why we want to use the eP (P-AET) rather than P or AET individually, because of the more direct connections between eP and water yield revealed by the water balance equation ($WY = P - AET + \Delta S$). Now, we have rephrased related texts to describe the selection of climate and vegetation index for building the DMC model in Line 111-117.

**Line 112-118:** The selection of climate and vegetation indices used in the DMC technique is an important issue. Previous studies have shown that effective precipitation (eP, P-AET) can reflect more information of climate on WY compared with individual P or AET, and be regarded as a reliable proxy to climate (Wei and Zhang, 2010; Zhang et al., 2019). LAI quantifies the amount of leaf area in an ecosystem and becomes an important variable reflecting vegetation structures and biophysical processes (Fang et al., 2019; Forzieri et al., 2020), and Li et al. (2021) has used LAI to investigate vegetation effects on seasonal hydrology in the UBR basin. Hence, we consider eP and LAI as the indices of climate and vegetation respectively, and use their time series as the inputs in the DMC model.

**Comment 0.7:** L172: Did the authors take the sum of the mean annual LAI or did they compute the cumulative in a different way? How does LAI vary spatially and temporally in each catchment? Please add more details.

**Reply:** We always use the cumulative LAI (or eP) as the explained variable and the cumulative WY as the response variable to build the double cumulative curve. The calculation of the cumulative values is the same for ep, LAI, and WY by taking the sum of the mean annual values. In the revised version, we provided a schematic diagram to show the DMC's procedure (Figure 2).

We also added the time-series of LAI in individual basins in the supplementary (Figure S2)

**Comment 0.8:** In the results section, please remove all sentences referring to the discussion of the main findings.

**Reply:** Thanks for your suggestions. We have removed them into discussion.

**Comment 0.9:** Section 4: I would like to see in this section also a discussion of the limitations of the approach used in this study. For instance, what is the uncertainty in precipitation, actual evapotranspiration and LAI in these catchments, and how do the uncertainties affect the main findings of this manuscript?

**Reply:** Thanks for your suggestions. We have rewritten the discussion, including (1) uncertainties and limitations of the used data and DMC method used here, and (2) provided broad implications for mountainous water resource management.

The limitations about the DMC model used in the study are presented in Line 234-248

**Line 235-249:** This study has some limitations regarding the DMC model to partition the effects of climatic and cryospheric changes on the hydrological regime shifts in the UBR basin. The DMC method is a useful alternative statistical method to physical modeling approaches, especially in alpine river basins (e.g. UBR basin) where there is less knowledge on the complex hydrological

mechanisms. While the method is still dependent on our prior understanding of hydrological responses to warming and related environmental changes, such as glaciers melting and vegetation greening. For the UBR basin, besides climate change, cryosphere (Biemans et al., 2019; Yao et al., 2019) and vegetation (Li et al., 2021, 2019a) are two major factors for hydrological changes, and the cryospheric contributions can be regarded as the deviations between total water yield and climate and vegetation contributions estimated by the DMC method. While, in some mountainous basins, human activities, such as urbanization, dam regulation and irrigation, may consume severely water resource or change seasonal runoff patterns, and thus we have to consider anthropogenic impacts into the DMC statistical model for river flow attribution. On the other hand, the DMC method applies the linear assumption between two variables, and thus it may fail to capture some nonlinear process among the interactions among water yield, vegetation and cryospheric melting in the study. Thus, with the availability of long-term in-situ observations and high-resolution remote sensing datasets in the UBR basin (Wang et al., 2022), other powerful statistical models considering nonlinear and casual structures should be applied to identify the causes of water yield changes (Runge et al., 2019)

The uncertainties about the data used in the study are presented in Line 249-262

**Line 250-263:** The data used in the study may also give rise to some uncertainties for our results. The 10x10 km precipitation product used here is generated by topographical and linear corrections based on observations. As Sun and Su (2020) pointed, while, the results of the linear correction approach highly vary with the station density. For example, the increased numbers from 4 to 10 stations in the basin will decrease the mean annual precipitation by about 20 mm. Hence, the reconstruction of precipitation dataset will rely on the density of the observed stations. Besides the topographic correction, the effects of the basin size and climate seasonality should be considered in the work of precipitation reconstruction in the UBR basin due to the complex climate and environment (Sun et al., 2019). Compared with precipitation, the estimation of evaporation may be much more challenging in high mountains. Although GLEAM actual evaporation shows the good agreement with in-situ eddy covariance records (Yang et al., 2017), its model structure does not include wind speed and solar radiation, which may affect the estimation of sublimation, and thus total evaporation (Li et al., 2019b). In addition, the coarse spatial resolution with a $0.25°$ spatial resolution in GLEAM may be insufficient to estimate regional evaporation in the UBR basin. However, with the help of the Second Tibetan Plateau Scientific Expedition and Research, the observation networks in meteorology, cryosphere and hydrology will be built, which is expected to benefit reliable precipitation and evaporation estimation, and make developing physically-based cryosphere-hydrological modeling possible (Wang et al., 2022)

The implications of this study are presented in Line 264-274

**Line 264-274:** Understanding the hydrological regime shifts and their causes in the high mountains are especially important in managing water resources, especially balancing the co-benefits between mountains and downstream lowlands (Viviroli et al., 2011). In the study, the combined (offsetting or additive) effects from climate and cryosphere are detected (Figure 5), and further lead to either slight or substantial increases in WY in the entire UBR basin (Figure 4a). The combined effects often hinder the roles of each driver in hydrological changes (Wei et al., 2018; Zhang and Wei, 2021), which should be considered when designing water management strategies in the large transboundary river system. For example, the additive effect may be beneficial for mitigating droughts and water shortage during droughts, but it may exacerbate the flood risks due to increased precipitation

and accelerated melting of the cryosphere in the future (Immerzeel et al., 2013). In addition, our results clearly show that the melt waters from glaciers might have already surpassed the "peak water", and the associated hydrological changes will substantially affect future water resources management. Thus, the projections of the occurrence time of "peak water" will be important in managing mountainous water resources.

**Comment 0.10:** Fig. 1: In (a) it is difficult to identify the dots. In (b), the dots are small and partially hidden by the labels, whereas in (c) the labels of the histogram are too small. Furthermore, the terms 'unused land' and 'beach land' are unclear.

**Reply:** Thanks for your suggestions. We use the red point to indicate the hydrological stations in Figure 1a, and adjust the location of labels in Figure 1b. In addition, we placed the Figure 1c in the original manuscript into the supplementary (Figure S3), and added the percentage of snow and glaciers in Table 1 in the main text in the revised version.

We are so sorry for showing land cover classification in the second level, and now we only show the classification in the first level. That is, land cover includes cultivated land, forestland, grassland, water body, urban land, unused land, and glaciers and snow (see Figure S3).

[Figure]

Figure S3. The land use types in 2000 in the UBR basin.

In addition, the "unused land" in land cover classifications means "Those ecosystems in which less than one third of the area has vegetation or other cover". In general, Barren Land has thin soil, sand, or rocks. Barren lands include deserts, dry salt flats, beaches, sand dunes, exposed rock, strip mines, quaries, and gravel pits" (https://www.hq.nasa.gov/iwgsdi/Barren_Land.html)

**Comment 0.11:** Fig. 5 in the main manuscript and Fig. 6 in the response to the reviewers: Some grey/light brown regression lines seem very flat, but still the correlation coefficients are >0.5 and significant. Please check whether there are mistakes in the statistical analyses.

**Reply:** Thanks for your questions. We have checked the results and they are correct. Vegetation-induced water yield (tan points) is much weaker compared with those caused by climate and cryosphere, and hence the fitted line is significant but looks flattened when we showed all the data in each panel. To avoid some misunderstandings, we have labelled the Pearson's correlation coefficients and shown the regression line only when the correlation is significant ($p < 0.05$) in Figure 6 in the revised manuscript. In addition, we also take the UYZR basin as an example to only show vegetation contributions and water yield deviations here (Figure R2).

**Comment 0.12:** Table S2: Some turning points are not significant. Based on these results, DMC analyses should not be performed for those catchments where the turning points were not significant.

[Figure]

Figure 1. Location of (a) the Upper Brahmaputra River (UBR) basin in the Qinghai Tibet Plateau (QTP), which is from Li et al. (2021), and (b) six basins divided by Lhatse, Nugesha, Shigatse, Yangcun, Lhasa, and Nuxia hydrological stations.

[Figure]

Figure R2. The scatter plot between vegetation contributions ($\Delta WY_v$) and total water yield deviations ($\Delta WY_s$) in the UYZR basin.

**Reply:** Thanks for your question. As you pointed, the Pettitt method does not find the significant abruption in some watersheds (Table 1), which potentially indicates the divergent causes for hydrological changes in the UBR basin and the significance of comprehensive assessment on water yield responses to climate warming.

Although some watersheds don't have significant turning points, we still find hydrological regime shifts in these regions (Figure 4). For example, for the HYZR basin with non-significant turning point in 1995, water yield increased by ∼10%, and its changing direction is reversed from increasing to decreasing. The hydrological regime shifts in the entire UBR basin, including magnitude and direction, fuel our interest to find potential causes.

**Reviewer 1**

**Comment 1.1:** Li et al. provide a case study to study the influence of climate and cryosphere on the water yield in the Brahmaputra. To do this they collected a long time series for climatic and precipitation data and analyzed it to find the water yield has changed over the time period studied. They find that there are substantial changes and attribute this mainly to the combined effect of climate and cryosphere. I think this study has value (I especially like the introduction) and should be considered for publication. However, I have several issues that I think should be addressed.

**Reply:** We are grateful for Dr. Florian Ulrich Jehn's thoughtful evaluation and support of our work. We will reply to comments in the following, including **(1) clarifying the reasons for selections of climate and vegetation indices**, and **(2) providing a schematic diagram showing the procedure of the double mass curve (DMC)**.

**General Comments**

**Comment 1.2:** First, the study does not provide enough information about its data. For example, after reading the study I am still unsure what exactly is meant when the study talks about climate being a major factor in its analysis. Is it the mean temperature? Is it some indices? Is it something completely different? Same goes for the term cryosphere, which is used quite loosely.

**Reply:** We thank the reviewer for pointing this out. We now clearly define (1) the effective precipitation (eP, P-AET) as a proxy to assess climate effects on water yield, and (2) meltwaters from glaciers and snow under warming as the cryospheric component in this study.

Related definition about the selection of the climate indicator in the DMC is shown:

**Line 112–114:** The selection of climate and vegetation indices used in the DMC technique is an important issue. Previous studies have shown that effective precipitation (eP, P-AET) can reflect more information of climate on WY compared with individual P or AET, and be regarded as a reliable proxy to climate (Wei and Zhang, 2010; Zhang et al., 2019).

Related definition about meltwaters is shown:

**Line 106-108:** For the large and pristine UBR and other mountainous basins, climate, vegetation, and **cryosphere (melt waters from glaciers and snow under warming, see Biemans et al. 2019; Huss and Hock 2018)** play important roles in hydrology, and these three parts must be together considered to accurately estimate hydrological responses to warming.

**Line 119–121:** To obtain cryospheric contributions to WY, we firstly build two types of DMC plots (see Figure S4) to assess the contributions of climate (eP) and vegetation (LAI), and then subtract the sum of estimated contributions from total WY deviations as cryospheric effects (results are shown in Figure S5). The schematic diagram 2 and associated mathematical formulas are shown as follows: ...

**Comment 1.3:** Second, after reading the methods it is not clear to me how the study is able to differentiate between the influence on climate, cryosphere and vegetation. This section would

profit from a more in depth explanation. In addition, why using this method? Why do you think it is especially good for your kind of study?

**Reply:** We thank the reviewer for this question. In the revised version, we provide the detailed reasons and a schematic diagram (Figure 2) to show why and how we use the DMC in the study.

[Figure]

Figure 2. The schematic diagram showing how to estimate the effects of climate, vegetation, and cryosphere on water yield in the MYZR basin (Details in Methodology).

For a large natural mountainous watershed, it still remains not clear about the hydrological responses to climate change and associated environmental changes, e.g. vegetation and cryosphere. That leads to great uncertainties when assessing water yield changes using hydrological models. While, long-term annual runoff observations and high-resolution precipitation records in the UBR basin provide a good opportunity for statistical models to investigate hydrological responses to climate warming. The Double Mass Curve – a data-driven statistical model – has been widely

been applied to estimate water yield responses to environmental changes in the hydrological community. We assume that in the UBR basin, water yield is affected by climate (e.g. precipitation and evaporation), vegetation greening or browning, and cryospheric loss (e.g. glaciers and snow melting). Hence, we can estimate cryospheric contributions to water yield by subtracting the sum of contributions from climate and vegetation from total deviations using the DMC method. Related revisions are shown.

**Line 46–49:** ... Lastly, the present inadequate understanding of hydrological responses to complex interactions among climate, vegetation, and cryosphere limits the application of hydrological models in those glacier-fed watersheds (Pellicciotti et al., 2012). While, long-term observed runoff data and recent high-resolution precipitation records may give a pathway for using statistical methods to estimate runoff responses to warming in the UBR basin.

**Line 103-111:** The DMC used here is a plot of the cumulative data of one variable versus the cumulative data of another related variable in a concurrent period. It has previously been used to assess the individual effect of climate (Gao et al., 2011), forest disturbance (Wei and Zhang, 2010), wildfire (Hallema et al., 2018), and cryosphere (Brahney et al., 2017) on water resources. For the large and pristine UBR and other mountainous basins, climate, vegetation, and **cryosphere (melt waters from glaciers and snow under warming, see Biemans et al. 2019; Huss and Hock 2018)** play important roles in hydrology, and these three parts must be together considered to accurately estimate hydrological responses to warming. It is considerably hard to directly calculate the supply of melt waters to WY due to the lack of long-term glacier monitoring, while runoff observations and high-resolution climate and vegetation data make it possible to use the DMC technique, a data-driven statistical method, to estimate cryospheric contributions to WY.

In addition, with the help of the schematic diagram (Figure 2), we now clearly explain the procedure for using the DMC to separate climate, vegetation and cryosphere contributions to water yield (see details in Methodology).

**Comment 1.4:** Third, the study finds a turning point for the behavior of the river. This seems quite important to me, but is never really discussed. Why did this change happen? What consequences will it have?

**Reply:** We thank the reviewer for pointing this out, and have analyzed the causes of turning points and their implications in the revised manuscript.

In this study, climate changes and meltwaters from glaciers and snow mainly contribute to substantial water yield changes before and after the turning point. Figure 5 shows that the combined effects from climate and cryosphere contribute to magnitude increases in water yield, but Figure 6 shows that climate is more important for the trend changes in water yield, while melt waters have the potential to mitigate water shortage risks. Related revisions are shown:

**Line 180–183:** Climate and cryosphere – two important factors affecting WY – together contribute to over 80% average magnitude increases of WY in the entire UBR basin; however, they play both additive or offsetting roles (Figure 5), resulting in slight or substantial WY increases (Figure 4a).

**Line 190–193:** Results in Figure 6 show that, although the correlation varies greatly across basins ranging from 0.11 to 0.93 after the TP, climate typically is positively associated with total WY, in

which the correlation is significant in half of basins ($p < 0.05$), again revealing the major role of climate in the hydrological trends in the entire UBR basin.

**Line 197–201:** In contrast to positive contributions of climate, we find that WY caused by cryosphere exhibits a negative association with reduced total WY deviations in recent years in the UYZR ($r = -0.39$, $p > 0.05$) and LSR ($r = -0.36$, $p > 0.05$) basins. The negative but weak relationship indicates that melt waters from cryospheric loss may compensate for low flow, and even mitigate water shortage risks. Also, the compensating effect from cryosphere is much stronger in the MYZR ($r = 0.47$, $p > 0.05$), and together with climate contributions, contributes to the increasing WY trend (Figure 4).

Meanwhile, our results may suggest the occurrence time of "peak water" in the UBR basin, which is in line with Huss and Hock (2018), and we also add related analysis and discussion in the revised version.

**Line 201–207:** Different from other regions, however, the HYZR basin shows a significantly positive relationship between cryospheric contributions and total WY deviations ($r = 0.76$, $p < 0.05$), indicating that cryosphere instead of climate leads to the downward trend in headwaters. This signifies that in this region, cryospheric contributions have already passed a maximum supplying to river flow, due to decreased glaciers and snow under continuous warming. The is further verified by the relationship of cryospheric contributions to total WY ($RC_g$) with temperature (Figure S8). In the HYZR basin, WY resulting from the cryosphere continues to increase with temperature until a maximum is reached, beyond which cryospheric contribution to total WY begins to decrease.

**Line 227-233:** Cryospheric contribution is also important for water yield regime shifts – melt waters from glaciers and snow melting can alleviate water resources shortages, mainly caused by decreased precipitation in recent years (Figure 6+S7). This finding is also supported by observed glacier runoff data (Yao et al., 2010) and several modeling studies (Lutz et al., 2014; Zhang et al., 2020; Wang et al., 2021). However, after glacier runoff reaches a maximum, defined as "peak water" (Gleick and Palaniappan, 2010), cryospheric mass loss cannot sustain the rising melt waters with atmospheric warming (e.g. the HYZR basin in Figure S8), which is in agreement with Huss and Hock (2018).

In the revised manuscript, we add the implications of our results on water resource management in mountainous watersheds.

**Line 265–275:** Understanding the hydrological regime shifts and their causes in the high mountains are especially important in managing water resources, especially balancing the co-benefits between mountains and downstream lowlands (Viviroli et al., 2011). In the study, the combined (offsetting or additive) effects from climate and cryosphere are detected (Figure 5), and further lead to either slight or substantial increases in WY in the entire UBR basin (Figure 4a). The combined effects often hinder the roles of each driver in hydrological changes (Wei et al., 2018; Zhang and Wei, 2021), which should be considered when designing water management strategies in the large transboundary river system. For example, the additive effect may be beneficial for mitigating droughts and water shortage during droughts, but it may exacerbate the flood risks due to increased precipitation and accelerated melting of the cryosphere in the future (Immerzeel et al., 2013). In addition, our results clearly show that the melt waters from glaciers might have already surpassed the "peak water" (Figure S8), and the associated hydrological changes will substantially

affect future water resources management. Thus, the projections of the occurrence time of "peak water" will be important in managing mountainous water resources.

**Specific Comments**

**Comment 1.5:** The study states several times that the increase meltwater has the potential to alleviate the loss of water availability. I also think this is the case, but it should be made clearer that this will only be a temporary relief until the glaciers have melted.

**Reply:** We agree with the reviewer that meltwater is only a temporary relief, and have improved these statements in the revised version. As mentioned, there may be a "maximum cryospheric contribution to water yield" ('peak water', Gleick and Palaniappan 2010). Glacier runoff will increase with warming and compensate for low flow during droughts (see negative correlations with decrease runoff in most basins, Figure 6), while steadily decrease after reaching "peak water" due to the reduced glaciers and snow (see the positive correlation in the HYZR basin, Figure 6a).

In the revised manuscript, we analyze and discuss our results with "peak water". For example,

**Line 201–207:** Different from other regions, however, the HYZR basin shows a significantly positive relationship between cryospheric contributions and total WY deviations ($r = 0.76$, $p < 0.05$), indicating that cryosphere instead of climate leads to the downward trend in headwaters. This signifies that in this region, cryospheric contributions have already passed a maximum supplying to river flow, due to decreased glaciers and snow under continuous warming. The is further verified by the relationship of cryospheric contributions to total WY ($RC_g$) with temperature (Figure S8). In the HYZR basin, WY resulting from the cryosphere continues to increase with temperature until a maximum is reached, beyond which cryospheric contribution to total WY begins to decrease.

**Line 227–233:** Cryospheric contribution is also important for water yield regime shifts – melt waters from glaciers and snow melting can alleviate water resources shortages, mainly caused by decreased precipitation in recent years (Figure 6+S7). This finding is also supported by observed glacier runoff data (Yao et al., 2010) and several modeling studies (Lutz et al., 2014; Zhang et al., 2020; Wang et al., 2021). However, after glacier runoff reaches a maximum, defined as "peak water" (Gleick and Palaniappan, 2010), cryospheric mass loss cannot sustain the rising melt waters with atmospheric warming (e.g. the HYZR basin in this study), which is in agreement with Huss and Hock (2018).

**Comment 1.6:** What are the specific reasons that vegetation was studied? Are the any reasons to assume that the vegetation has changed significantly in the time period?

**Reply:** Thanks for your questions. We have added the reasons about vegetation effects on water yield in the revised manuscript.

Many studies have indicated that vegetation will significantly change water yield based on the statistical or physical models. Also, this recent study by Li et al. (2021) revealed that significant vegetation greening in this region may redistribute water resources through seasons in the UBR basin. Therefore, we build the DMC plot between cumulative LAI and WY to estimate vegetation effects on water yield. Related revisions are shown here:

**Line 33–36:** Vegetation has also been proven to be vital for mountainous water resources; Li et al. (2017) showed that evaporation, mostly due to grassland restoration, decreased WY in the Yangtze River basin, while Li et al. (2021) suggested that vegetation greening may change the seasonality of water resources and increase WY during the dry season in the UBR basin.

We also show the time series of LAI for individual basin in the supplementary (Figure S2), and find vegetation increases firstly and afterwards decrease in most basins.

[Figure]

Figure S2. The temporal changes of precipitation (P, blue line), actual evaporation (AET, orange line), and water yield (WY, green line) and Leaf Area Index (LAI, grey line) during 1982–2013 in the entire UBR basin. The vertical line indicates the turning point in WY.

**Comment 1.7:** Figure S2 belongs in the paper in my opinion, as it seems like this is your main plot, which all following plots refer to.

**Reply:** We thank the reviewer for this suggestion. In the revision, we place a schematic diagram (Figure 2) in main text, which will explain how to conduct the DMC analysis more clearly.

**Comment 1.8:** Figure 1: Please change this 3D pie chart to bar char, as those are much easier to read.

**Reply:** Thank you for pointing it out. We have placed Figure 1c into the supplementary (Figure S3), and provided detailed information in Table 1 in the revised manuscript.

**Comment 1.9:** Do the abbreviations that are used to label the subcatchments have any meaning?

[Figure]

Figure S3. The land use types in 2000 in the UBR basin.

**Reply:** Yes. For example, "HYZR" means the headwater watershed in the Yarlung Zangbo River (YZR) basin, and "LSR" means Lasha River basin. We add detailed introduction for abbreviations in Table 1 in main text.

Table 2: Information of six basins divided by the locations of hydrological stations. The column "TP" indicates the turning point using the Pettitt method, in which a significant tuning point is labeled with *. Glaciers and snow area is acquired from the land use and cover in 2000 (see details in Dataset).

| Abbrevation | Full names | Station | Total area (km$^2$) | Mean elevation (m) | Glaciers and snow percentage (%) | TP |
|---|---|---|---|---|---|---|
| HYZR | Headstream | Lhatse | 49,739 | 5061 | 1.7 | 1995 |
| UYZR | Upstream | Nugesha | 43,916 | 4985 | 0.39 | 1998* |
| NCR | Nianchu River | Shigatse | 14,359 | 4733 | 1.96 | 1997* |
| MYZR | Midstream | Yangcun | 20,004 | 4681 | 1.81 | 1997* |
| LSR | Lhasa River | Lhasa | 25,601 | 4879 | 0.72 | 1996 |
| LYZR | Downstream | Nuxia | 45,017 | 4586 | 2.51 | 1997 |

**Comment 1.10:** Did you check if you evapotransporiration is roughly correct? You used evapotransporation data from a global model, which might have not been calibrated well to regions such extreme as yours.

**Reply:** We thank the reviewer for pointing this out. As we do not have access to observed actual evaporation data in this region, we used AET from GLEAM, which has been extensively validated across varied vegetation types in China and has shown good performance with in situ observations (Yang et al., 2017).

**Line 83–85:** Regional actual evaporation (AET) with a 0.25° spatial resolution is acquired from Global Land Evaporation Amsterdam Model (GLEAM) (Martens et al., 2017). The evaporation product has been validated in different biome types in China and shown high correlations with in-situ eddy covariance AET (Yang et al., 2017).

Also, we discuss data uncertainties in the revised version.

**Line 256–263:** Compared with precipitation, the estimation of evaporation may be much more challenging in high mountains. Although GLEAM actual evaporation shows the good agreement with in-situ eddy covariance records (Yang et al., 2017), its model structure does not include wind

speed and solar radiation, which may affect the estimation of sublimation, and thus total evaporation (Li et al., 2019b). In addition, the coarse spatial resolution with a 0.25° spatial resolution in GLEAM may be insufficient to estimate regional evaporation in the UBR basin. However, with the help of the Second Tibetan Plateau Scientific Expedition and Research, the observation networks in meteorology, cryosphere and hydrology will be built, which is expected to benefit reliable precipitation and evaporation estimation, and make developing physically-based cryosphere-hydrological modeling possible (Wang et al., 2022).

**Comment 1.11:** Why did you choose LAI as a proxy for vegetation and not some other measure?

**Reply:** Thanks for your questions about the selections of vegetation indices. We give the reasons in the revised version.

**Line 114–117:** LAI quantifies the amount of leaf area in an ecosystem and becomes an important variable reflecting vegetation structures and biophysical processes (Fang et al., 2019; Forzieri et al., 2020), and Li et al. (2021) has used LAI to investigate vegetation effects on seasonal hydrology in the UBR basin.

**Comment 1.12:** Have you considered also checking for the runoff-ratio? This seems like a variable that should give you some additional information.

**Reply:** We thank the reviewer for this suggestion. As effective precipitation is one of the main explanatory variable considered in this study, we decided to use just water yield, or runoff depth as the target variable. This allows us to robustly apply the statistical method (DMC) to quantify hydrological responses to climate warming.

**Comment 1.13:** Please change Fig. 3 and Fig 4. to boxplots or swarmplots (depending on your sample size you calculated your mean and standard deviation from). Having just a bar plot with a standard deviation does not really show how your underlying data looks like.

**Reply:** We agree with the reviewer. We have changed the two figures to boxplots to support our analysis. Note the labelled numbers have been changed (Figure 4 and 5).

[Figure]

Figure 4. Water yield regime shifts in the entire UBR basin. (a) Magnitude of water yield changes. Black "x" signals show the mean of water yield in each boxplot. (b) Direction of water yield changes. The black hatching represents the statistically significant trend ($p < 0.05$). The color of boxes represents the period before (light color) and after (dark color) the turning point (TP).

**Comment 1.14:** Are your p-values corrected? If not, this would mean that likely in Figure 5 there are way fewer significant trends.

[Figure]

Figure 5. Attribution analysis of magnitude increases in water yield due to climate ($\Delta WY_c$, blue box), vegetation ($\Delta WY_v$, tan box), and cryosphere ($\Delta WY_g$, red box), and their relative contributions (the bar with colors on the top) in each basin. Black "x" signals show the mean of water yield deviations (see Figure S5) in each boxplot.

**Reply:** Thanks for your comments. The p value here is correct, but I am sorry that the labelled $n$ is wrong due to a minor programming error. In the revised version, we have corrected this error, and will only show the regression lines in Figure 6 when p value is less than 0.05, and use "significant" or "significantly" in main text. For example:

**Line 191-192:** ..., climate typically is positively associated with total WY, in which the correlation is significant in half of basins ($p < 0.05$), ...

**Line 198-199:** The negative butweak relationship indicates that melt waters from cryospheric loss may compensate for low flow, and even mitigate water shortage risks.

**Comment 1.15:** The text is quite heavy on abbreviations, which makes it harder to read. Please consider just writing the words out instead of abbreviating them.

**Reply:** Thank you for the suggestions. We have deleted some abbreviations in main text, such as "Third Pole (TP)". And we also provide a table (see Table 1 in main text) to indicate some abbreviations clearly.

**Technical corrections**

**Comment 1.16:** L19-21: I am not able to parse this sentence.

**Reply:** We thank the reviewer for pointing this out. We have rewritten the abstract and convey the main information clearer to readers.

[revised manuscript text omitted]

**Reviewer 2**

**Comment 2.1:** The manuscript from Li et al. offers an analysis of the different drivers of hydrological regime shifts in the upper Brahmaputra, highlighting changing climate and glacier loss as major determinants. While the work is generally well written and shaped, there are some issues/suggestions to be considered before publication:

**Reply:** We thank the reviewer for their constructive comments and specifically for the suggestions on using the idea of "peak water" as a conceptual framework for understanding cryospheric contributions to river flow. **We will clarify some definitions, and give related evidence with "peak water" in Results and Discussion.**.

**General Comments**

**Comment 2.2:** A conceptual framework would greatly help you to shape the storyline. Actually, there are several studies showing regime shifts associated with glacier loss (see Huss & Hoch, 2018; concept of "peak water"). I think that your work provides further evidence in that direction, showing the magnitude of hydrological glacier influence over spatial gradients, and offering an important analysis of the turning points in regime shifts.

**Reply:** We thank the reviewer for suggestions. We have now framed our analysis and discussion within the conceptual framework of "peak water" as you suggested. For example,

**Line 197–209:** In contrast to positive contributions of climate, we find that WY caused by cryosphere exhibits a negative association with reduced total WY deviations in recent years in the UYZR (r = -0.39, p > 0.05) and LSR (r = -0.36, p > 0.05) basins. The negative but weak relationship indicates that melt waters from cryospheric loss may compensate for low flow, and even mitigate water shortage risks. Also, the compensating effect from cryosphere is much stronger in the MYZR (r = 0.47, p > 0.05), and together with climate contributions, contributes to the increasing WY trend (Figure 4). Different from other regions, however, the HYZR basin shows a significantly positive relationship between cryospheric contributions and total WY deviations (r = 0.76, p < 0.05), indicating that cryosphere instead of climate leads to the downward trend in headwaters. This signifies that in this region, cryospheric contributions have already passed a maximum supplying to river flow, due to decreased glaciers and snow under continuous warming. The is further verified by the relationship of cryospheric contributions to total WY ($RC_g$) with temperature (Figure S8). In the HYZR basin, WY resulting from the cryosphere continues to increase with temperature until a maximum is reached, beyond which cryospheric contribution to total WY begins to decrease. In addition, the compensating effect of melt waters can be seen clearly in the UYZR, MYZR and LSR basins, i.e., WY caused by cryospheric loss keeps a positive relationship with the increase of temperature, further supporting the higher correlation in these basins (Figure 6).

**Line 227–233:** Cryospheric contribution is also important for water yield regime shifts – melt waters from glaciers and snow melting can alleviate water resources shortages, mainly caused by decreased precipitation in recent years (Figure 6+S7). This finding is also supported by observed glacier runoff data (Yao et al., 2010) and several modeling studies (Lutz et al., 2014; Zhang et al., 2020; Wang et al., 2021). However, after glacier runoff reaches a maximum, defined as "peak water" (Gleick and Palaniappan, 2010), cryospheric mass loss cannot sustain the rising melt waters with

atmospheric warming (e.g. the HYZR basin in Figure S8), which is in agreement with Huss and Hock (2018).

[Figure]

Figure S8. The relationship between cryospheric contributions to water yield deviations ($RC_g(t)$, see Methods in main text) and annual mean 2m maximum air temperature ($T_{max}$) using the polynomial fitting. The colorbar indicates the years after a TP in individual basins. R square here used to evaluate the fitting goodness is labelled in each panel.

**Comment 2.3:** I found the discussion a little bit lacking. As glaciers ad the climate are the major hydrological drivers, what will it happen when glaciers disappear and the climate has shifted? Also, you should stress that the "climate shifts" are changes in precipitation patterns in your work.

**Reply:** We thank the reviewer for pointing this out. We have added the detailed analysis and discussions within the framework of "peak water".

**Line 201–207:** Different from other regions, however, the HYZR basin shows a significantly positive relationship between cryospheric contributions and total WY deviations (r = 0.76, p < 0.05), indicating that cryosphere instead of climate leads to the downward trend in headwaters. This signifies that in this region, cryospheric contributions have already passed a maximum supplying to river flow, due to decreased glaciers and snow under continuous warming. The is further verified by the relationship of cryospheric contributions to total WY ($RC_g$) with temperature (Figure S8). In the HYZR basin, WY resulting from the cryosphere continues to increase with temperature until a maximum is reached, beyond which cryospheric contribution to total WY begins to decrease.

**Line 227–233:** Cryospheric contribution is also important for water yield regime shifts – melt waters from glaciers and snow melting can alleviate water resources shortages, mainly caused by decreased precipitation in recent years (Figure 6+S7). This finding is also supported by observed glacier runoff data (Yao et al., 2010) and several modeling studies (Lutz et al., 2014; Zhang et al.,

2020; Wang et al., 2021). However, after glacier runoff reaches a maximum, defined as "peak water" (Gleick and Palaniappan, 2010), cryospheric mass loss cannot sustain the rising melt waters with atmospheric warming (e.g. the HYZR basin in Figure S8), which is in agreement with Huss and Hock (2018).

In the revised manuscript, we also stress the importance of precipitation on water yield.

**Line 225–227:** Climate, especially precipitation, still control the declining WY trend after the TP in most regions (Figure 6 and S7), may become an important factor in occurrence of turning points (Figure 3c+d). This suggests the importance of precipitation and its projections on future hydrological process in mountainous watersheds (Lutz et al., 2014).

[Figure]

Figure S7. Direction of precipitation (a) and actual evaporation (b) changes. The black hatching represents the statistically significant trend (p < 0.05). The color of boxes represents the period before (light color) and after (dark color) the turning point (TP).

**Comment 2.4:** Why did you choose this particular type of analysis to estimate drivers of regime changes? This is not sufficiently explained in the text

**Reply:** Thank you for your suggestions. We have supplemented reasons and an example (Figure 2) for the use of DMC method in the Introduction and Methodology sections.

**Line 46–50:** Lastly, the present inadequate understanding of hydrological responses to complex interactions among climate, vegetation, and cryosphere limits the application of hydrological models in these mountainous watersheds (Pellicciotti et al., 2012). While, long-term observed runoff records and recent high-resolution precipitation datasets give a pathway for using statistical methods to estimate runoff responses to warming in the UBR basin.

**Line 103–111:** The DMC used here is a plot of the cumulative data of one variable versus the cumulative data of another related variable in a concurrent period. It has previously been used to assess the individual effect of climate (Gao et al., 2011), forest disturbance (Wei and Zhang, 2010), wildfire (Hallema et al., 2018), and cryosphere (Brahney et al., 2017) on water resources. For the large and pristine UBR and other mountainous basins, climate, vegetation, and cryosphere (melt waters from glaciers and snow under warming, see Biemans et al. 2019; Huss and Hock 2018) play important roles in hydrology, and these three parts must be together considered to accurately estimate hydrological responses to warming. It is considerably hard to directly calculate the supply of melt waters to WY due to the lack of long-term glacier monitoring, while runoff observations and high-resolution climate and vegetation data make it possible to use the DMC technique, a data-driven statistical method, to estimate cryospheric contributions to WY.

[Figure]

Figure 2. The schematic diagram showing how to estimate the effects of climate, vegetation, and cryosphere on water yield in the MYZR basin (Details in Methodology).

**Line 112–118:** The selection of climate and vegetation indices used in the DMC technique is an important issue. Previous studies have shown that effective precipitation (eP, P-AET) can reflect more information of climate on WY compared with individual P or AET, and be regarded as a reliable proxy to climate (Wei and Zhang, 2010; Zhang et al., 2019). LAI quantifies the amount of leaf area in an ecosystem and becomes an important variable reflecting vegetation structures and biophysical processes (Fang et al., 2019; Forzieri et al., 2020), and Li et al. (2021) has used LAI to investigate vegetation effects on seasonal hydrology in the UBR basin. Hence, we consider eP and LAI as the indices of climate and vegetation respectively, and use their time series as the inputs in the DMC model.

**Line 119–122:** To obtain cryospheric contributions to WY, we firstly build two types of DMC plots (see Figure S4) to assess the contributions of climate (eP) and vegetation (LAI), and then subtract the sum of estimated contributions from total WY deviations as cryospheric effects (results are shown in Figure S5). The schematic diagram 2 and associated mathematical formulas are shown as follows:

**Comment 2.5:** It seems that the drivers of regime shifts depend on the considered part of the

catchment. In general, the influence from glaciers is higher in the upper part and that from precipitation is higher at downstream locations. I think that translating this information into "spatial gradients/turning points" would greatly improve the quality of your work. Is there any relationship between the glacier cover in the catchment and the role of glaciers in driving the magnitude and direction of regime shifts associated with glacier loss or precipitation changes?

**Reply:** The turning point is both controlled by climate and cryospheric loss, and thus it may be not possible to directly build relationships between turning points and cryospheric contributions. In addition, the limited data (we only access snow and glacier area in 2000) may hinder the analysis between glacier areas and cryospheric contributions from the DMC method.

[Figure]

Figure S8. The relationship between cryospheric contributions to water yield deviations ($\Delta WY_s(t)$) and annual mean 2m maximum air temperature ($T_{max}$) using the polynomial fitting. R square used to evaluate the fitting goodness is labelled in each panel.

But, Figure 1 in Huss and Hock (2018) recommended by the reviewer shows the responses of cryospheric contributions to river flow under global warming. Based on this, we try to link cryospheric contributions with temperature changes, and find a nonlinear relationship between them (see Figure S8). Related revisions are described here.

**Line 227–233:** Cryospheric contribution is also important for water yield regime shifts – melt waters from glaciers and snow melting can alleviate water resources shortages, mainly caused by decreased precipitation in recent years (Figure 6+S7). This finding is also supported by observed glacier runoff data (Yao et al., 2010) and several modeling studies (Lutz et al., 2014; Zhang et al., 2020; Wang et al., 2021). However, after glacier runoff reaches a maximum, defined as "peak water" (Gleick and Palaniappan, 2010), cryospheric mass loss cannot sustain the rising melt waters with atmospheric warming (e.g. the HYZR basin in Figure S8), which is in agreement with Huss and Hock (2018).

**Line 272–275:** In addition, our results clearly show that the melt waters from glaciers might have already surpassed the "peak water" (Figure S8), and the associated hydrological changes will substantially affect future water resources management. Thus, the projections of the occurrence time of "peak water" will be important in managing mountainous water resources.

**Abstract**

**Comment 2.6:** I suggest you to remove useless adverbs such as "however, nevertheless, etc.". Try to shorten the abstract a little bit, e.g., discarding not essential sentences or summarizing some concepts

**Reply:** Thanks. We have rewritten the abstract (see below).

**Comment 2.7:** Line 15. Change "melt" with "loss". Cryospheric changes can increase the amount of available water, e.g. in rock glaciers

**Reply:** Thanks. We have rewritten the abstract (see below).

**Comment 2.8:** Line 18. Is it "stream head" correct word? I would delete the part ", as represented...downstream" in this sentence, useless for the abstract in my opinion.

**Reply:** Thanks. We have deleted the useless sentence for the abstract to ensure it convey main information to readers (see below).

**Comment 2.9:** Line 19. Delete "we found that"

**Reply:** Thanks. We have rewritten the abstract (see below).

**Comment 2.10:** Line 21. Delete "furthermore"

**Reply:** Thanks. We have rewritten the abstract (see below).

**Comment 2.11:** Line 23. Delete "however"

**Reply:** Thanks. We have deleted it (see below).

**Comment 2.12:** Line 25. Delete "nevertheless, we found that"

**Reply:** Thanks. We have rewritten the abstract to strengthen its coherence (see below).

**Comment 2.13:** Line 28. What do you mean with "ecological restoration"? I would remove this word as "water management" is enough in this context. Either, you can use the word "water governance", which involves the management of water as well as the related ecosystems and resources

**Reply:** Thanks for your advice. We deleted "ecological restoration" throughout the manuscript (see below).

The revised abstract is as follows:

**Line 1–12:** Although evidence of hydrological responses to climate is abundant, the reliable assessments of water yield (WY) over mountainous regions, such as the Upper Brahmaputra River (UBR) basin, remain unclear due to intensified cryospheric changes. Based on multi-station runoff observations, we examine long-term WY changes during 1982–2013 in the UBR basin, and find there are in general hydrological regime shifts in the late 1990s; magnitude increases in WY range from ∼10% to ∼80%, while its directions reverse from upward to downward after the late 1990s. Then, the double mass curve (DMC) technique is used to assess the effects of climate, vegetation, and cryosphere on WY changes. Results show that climate and cryosphere together contribute to over 80% of magnitude increases of WY in the entire UBR basin, in which the role of vegetation is nearly negligible. The combined effects, however, are either offsetting or additive, leading to slight or substantial magnitude increases, respectively. Climate change, particularly precipitation decrease leads to the downward WY trend in recent years, while melt waters under global warming may alleviate the water shortage in some basins. Therefore, the combined effects of climate and cryosphere on WY should be considered in future water resources management over mountainous basins, particularly involving co-benefits between upstream and downstream regions.

**Introduction**

**Comment 2.14:** Line 36. What do you mean with glacial snowmelt? Cryospheric drivers are snow and glaciers providing water across the melting process, i.e., glacier ice melt and snowmelt. Or do you mean the snowmelt occurring on the glacier surface? Please consider here the paper from Huss et al., 2017, which also includes the permafrost ice as a key component of the mountain cryosphere

**Reply:** Thank you very much for pointing it out. We are so sorry for the misunderstanding and we use **"glaciers and snow melting"** throughout the revised manuscript.

**Comment 2.15:** Line 36. It is actually unclear to a layman what the Third Pole is. Please clearly and concisely define it the first time you name it

**Reply:** Thanks. "Third pole" has the less role in the manuscript and also cause the confusion to the abbreviation of "turning point", so we have deleted it.

**Comment 2.16:** Line 52. "direction of change..." of what?

**Reply:** Thanks. We have used the following expression to replace it:

**Line 29–30:** For example, Fan and He (2015) highlighted the important role of precipitation in WY increases in the Salween and Mekong River basins.

**Comment 2.17:** Line 54. "glacial snow". See same comment of line 36.

**Reply:** Thanks. We have used **"glaciers and snow melting"** throughout the manuscript.

**Comment 2.18:** Line 84-85. "a reference...modelling". You already provided this sentence 8 lines earlier. Please avoid repetition.

**Reply:** Thanks. We have deleted the repeated expressions.

**Results**

**Comment 2.19:** Figure 3. I think the use of boxplots would greatly help interpretation.

**Reply:** We have changed the figure to the boxplots to show the data distribution (Figure 4).

**Comment 2.20:** Figure 5. I suggest you to provide the text and fitting lines for significant relationships only, to avoid figure overwhelming and help interpretation

**Reply:** Thanks for your suggestions. We have created the figure to avoid overwhelming as following.

[Figure]

Figure 4. Water yield regime shifts in the entire UBR basin. (a) Magnitude of water yield changes. Black "x" signals show the mean of water yield in each boxplot. (b) Direction of water yield changes. The black hatching represents the statistically significant trend (p < 0.05). The color of boxes represents the period before (light color) and after (dark color) the turning point (TP).

[Figure]

Figure 6. The correlation between time series of total water yield deviation ($\Delta WY_s(t)$, x-axis) and its components (y-axis) caused by climate ($\Delta WY_c(t)$, blue point), vegetation ($\Delta WY_v(t)$, tan point), and cryosphere ($\Delta WY_g(t)$, red point), respectively. The fitting line and its 95% confidence interval are shown only when p value < 0.05. $n$ indicates the number of years after the TP, which is determined by the Pettitt method (See Table 1 and Figure 3c).

**Discussion**

**Comment 2.21:** I think that an important work to be considered, that may help contextualising your

storyline, would be Huss & Hock (2018) providing the concept of "peak water". I suggest you to reshape the discussion around this work. Your results clearly show that the upper Brahmaputra has already surpassed the Peak Water and is now in declining phase of hydrological changes associated with glacier loss. This conceptualisation would also help you to better discuss the turning points that different areas experienced during distinct years…the presence of these turning points should be better discussed, as it is one of the strengths of the chosen methodology.

**Reply:** Thank you very much for your constructive comments. We have linked the results with "peak water", as revealed in (Huss and Hock, 2018).

**Line 197–209:** In contrast to positive contributions of climate, we find that WY caused by cryosphere exhibits a negative association with reduced total WY deviations in recent years in the UYZR (r = -0.39, p > 0.05) and LSR (r = -0.36, p > 0.05) basins. The negative but weak relationship indicates that melt waters from cryospheric loss may compensate for low flow, and even mitigate water shortage risks. Also, the compensating effect from cryosphere is much stronger in the MYZR (r = 0.47, p > 0.05), and together with climate contributions, contributes to the increasing WY trend (Figure 4). Different from other regions, however, the HYZR basin shows a significantly positive relationship between cryospheric contributions and total WY deviations (r = 0.76, p < 0.05), indicating that cryosphere instead of climate leads to the downward trend in headwaters. This signifies that in this region, cryospheric contributions have already passed a maximum supplying to river flow, due to decreased glaciers and snow under continuous warming. The is further verified by the relationship of cryospheric contributions to total WY ($RC_g$) with temperature (Figure S8). In the HYZR basin, WY resulting from the cryosphere continues to increase with temperature until a maximum is reached, beyond which cryospheric contribution to total WY begins to decrease. In addition, the compensating effect of melt waters can be seen clearly in the UYZR, MYZR and LSR basins, i.e., WY caused by cryospheric loss keeps a positive relationship with the increase of temperature, further supporting the higher correlation in these basins (Figure 6).

**Line 227–233:** Cryospheric contribution is also important for water yield regime shifts – melt waters from glaciers and snow melting can alleviate water resources shortages, mainly caused by decreased precipitation in recent years (Figure 6+S7). This finding is also supported by observed glacier runoff data (Yao et al., 2010) and several modeling studies (Lutz et al., 2014; Zhang et al., 2020; Wang et al., 2021). However, after glacier runoff reaches a maximum, defined as "peak water" (Gleick and Palaniappan, 2010), cryospheric mass loss cannot sustain the rising melt waters with atmospheric warming (e.g. the HYZR basin in this study), which is in agreement with Huss and Hock (2018).

**Line 265–275:** Understanding the hydrological regime shifts and their causes in the high mountains are especially important in managing water resources, especially balancing the co-benefits between mountains and downstream lowlands (Viviroli et al., 2011). In the study, the combined (offsetting or additive) effects from climate and cryosphere are detected (Figure 5), and further lead to either slight or substantial increases in WY in the entire UBR basin (Figure 4a). The combined effects often hinder the roles of each driver in hydrological changes (Wei et al., 2018; Zhang and Wei, 2021), which should be considered when designing water management strategies in the large transboundary river system. For example, the additive effect may be beneficial for mitigating droughts and water shortage during droughts, but it may exacerbate the flood risks due to increased precipitation and accelerated melting of the cryosphere in the future (Immerzeel et al., 2013). In addition, our results clearly show that the melt waters from glaciers might have already

surpassed the "peak water" (Figure S8), and the associated hydrological changes will substantially affect future water resources management. Thus, the projections of the occurrence time of "peak water" will be important in managing mountainous water resources.

**Comment 2.22:** Line 309-311. This is not true! See works from Huss and Hoch (2018), and in general the latest IPCC report on the ocean and the cryosphere, chapter dedicated to mountain environments...

**Reply:** Thank you very much for pointing it out. In the original manuscript, we want to show that this study provides a more detailed analysis for water yield changes in the UBR basin. We have deleted it to avoid the confusion.

**Comment 2.23:** Line 330. Please change "retractration" with "loss" or "recession". Also the sentence is unclear as written

**Reply:** Thanks. We have rewritten this sentence.

**Line 228–230:** ..., melt waters from glaciers and snow melting can alleviate water resources shortages, ... This finding is also supported by observed glacier runoff data (Yao et al., 2010) and several modeling studies (Lutz et al., 2014; Zhang et al., 2020; Wang et al., 2021).

**Comment 2.24:** Line 338. What do you mean with "ecological restoration"? It is unclear why this would help ecological restoration in particular. It is a general issue of water governance, after all, not just restricted to ecological restoration.

**Reply:** Thank you very much for your comment. We use "water resources management" to indicate the implication of this study throughout the main text.

---

## Author Response (AR2)

Response to Editors' and Reviewers' Comments: **Climate and cryosphere cause water yield regime shifts in the Upper Brahmaputra River basin**

Dear Dr. Giulia Zuecco,

We thank you for giving the opportunity to submit a revised manuscript to the Journal *Hydrology and Earth System Sciences*. We have carefully considered and addressed the comments from you and two reviewers. In doing so, we believe that the quality of the manuscript has increased substantially. We hope that the revised manuscript has satisfactorily addressed the comments and questions in the previous round of review.

In the revised version, we have (1) clearly presented the definitions of various terms used throughout the study, (2) rephrased the equations and relevant descriptions, and (3) made the Python codes used for data analysis and plotting figures publicly available ([https://github.ugent.be/haohaoli/HESS-2022-Water-Yield.git](https://github.ugent.be/haohaoli/HESS-2022-Water-Yield.git)).
Below we provide a point-by-point response to the comments and concerns raised by the editor and reviewers. All modifications in the manuscript have been marked.

Sincerely

Hao Li (on behalf of all coauthors)
PhD candidate, [Hao.liwork@ugent.be](mailto:Hao.liwork@ugent.be)
Hydro-Climate Extremes Lab, Ghent University
Sep 27, 2022

Li Hao

**Editor**

**Comment 0.1:** Thank you for considering my and the reviewers' comments, and resubmitting an improved version of the manuscript. However, I think you still need to clarify the use of some terms and the methodological approach, and to improve the discussion (please see the comments of the two reviewers). Particularly, I invite you to carefully consider the comments of the first reviewer about the uncertainty affecting your dataset and data availability. In addition, please see the following comments (lines refer to the manuscript without track changes).

**Reply:** We thank the editor and the encouraging assessment of our revised study. In this round of revision, we have added the terms' definiation in Introduction, clarified the used method in Data and Methods, discussed the uncertainties in LAI, and incorporated your suggestions into the revised manuscript. In addition, we have made the python scripts available through an open code repository based on the comments from Dr. Florian Ulrich Jehn (`https://github.ugent.be/haohaoli/HESS-2022-Water-Yield.git`).

**Comment 0.2:** Introduction: the terms, climate, cryosphere and vegetation, have not been clarified. Please explain them.

**Reply:** We thank Dr. Giulia Zuecco for pointing this out. In the revised manuscript, we define these key terms in the last paragraph in Introduction, which helps readers understand this study clearly.

**Line 51–54:** And then, we use DMC method to estimate the effects of climate that is indicated by effective precipitation (P-E, eP), vegetation (represented by Leaf Area Index, LAI), and cryosphere (e.g. melt waters from glaciers and snow) on magnitude and direction changes in WY (See Data and Methods).

In addition, we have rephrased the Introduction part to avoid introducing these terms when we introduce other relevant studies. For example,

**Line 26–27:** Therefore, comprehensively assessing long-term changes of WY, particularly magnitude and direction, is of great importance for the sustainable development of water resources in the QTP (Yao et al., 2019).

**Line 29–30:** In recent years, WY has been significantly affected by multiply factors in the QTP.

**Line 45–48:** Lastly, the present inadequate understanding of hydrological responses to complex interactions among multi-spheres limits the application of hydrological models in these mountainous watersheds (Pellicciotti et al., 2012).

**Comment 0.3:** L83-85: Please report the exact GLEAM products and GLEAM version that were used for the data collection and analysis. Furthermore, please check the use of the term "evaporation" because in the supplementary material you used "evapotranspiration" (caption of Fig. S1).

**Reply:** Thank you for pointing this out. We have reported the version of the GLEAM used in our study. Also, we always keep "evaporation" or "E" in the revised version.

**Line 84–85:** The evaporation (E) with a 0.25° spatial resolution is acquired from Global Land Evaporation Amsterdam Model (GLEAM) **version 3.5a**.

**Comment 0.4:** Table 1: Unclear use of the terms "glaciers and snow percentage".

**Reply:** We thank the editor for pointing this out. We have added the column "area" to indicate the actual area of glaciers and snow, and another column "percent" to show its percentage of total area in individual basins.

Table 1: Information of six basins divided by the locations of hydrological stations. The column "Tp" indicates the turning point using the Pettitt method, in which a significant turning point is labeled with *. Glaciers and snow is acquired from the land use and cover in 2000 (see Data). The unit of area is km$^2$, and the unit of elevation is m.

| Abbreviation | Full names | Station | Total area | Mean elevation | Glaciers and snow | | Tp |
| --- | --- | --- | --- | --- | --- | --- | --- |
| | | | | | area | percent | |
| HYZR | Headstream | Lhatse | 49,739 | 5,061 | 853 | 1.71 | 1995 |
| UYZR | Upstream | Nugesha | 43,916 | 4,985 | 175 | 0.40 | 1998* |
| NCR | Nianchu River | Shigatse | 14,359 | 4,733 | 282 | 1.96 | 1997* |
| MYZR | Midstream | Yangcun | 20,004 | 4,681 | 360 | 1.80 | 1997* |
| LSR | Lhasa River | Lhasa | 25,601 | 4,879 | 185 | 0.72 | 1996 |
| LYZR | Downstream | Nuxia | 38,419 | 4,586 | 963 | 2.51 | 1997 |

**Comment 0.5:** I think you should keep glacier and snow covers separated. For instance, you could quantify the glacier cover during the ablation season, and clearly describe in the text when the seasonal snowpack (outside the glacierized area) is present and at which elevations.

**Reply:** Thanks for your suggestions. We agree that separating the effects of glaciers and snow melt is interesting. However, it is not trivial using the statistical techniques employed in this study. We use a statistical method (DMC) to isolate the effects of climate and vegetation on water yield. The remaining part from total water yield deviations is regarded as the contributions from cryosphere changes, which includes both changes in glacier and seasonal snow melt.

In this study, we present the total area and percentage of glaciers and snow from the land use cover map in 2000 (Figure S3 and Table 1, which supports the truth that the UBR basin is covered by glaciers and snow (Bibi et al., 2018). Hence, under global warming, this region is experiencing significant changes in cryosphere and having impacts on mountainous water resources, which fuels my interest in detecting hydrological changes and their causes.

**Comment 0.6:** Section 2.3.2: Please explain all subscripts used for WY. For equation 1, please describe the values that tp can take and check WYo(t), which is reported twice (based on the text above, I think it is not correct). The description for computing WYc and WYv is unclear; please rephrase and report the equations. Please always use capital (or small) letters for tp throughout the manuscript.

**Reply:** Thank you for pointing them out. We hope to provide a general framework for six basins by using equations and relevant statements, so we always use "Tp" to indicate the turning point. Also, we have made sure that the term $WY_o(t)$ represents water yield from observations in the

texts and equations. To state these equations clearly in the revised version, we firstly define a variable $\overline{WY_{ob}}$ that depicts the averaged water yield before Tp, which will reduce the numbers of variables within equations. And then, we add the range of Tp in equations, e.g. $t = Tp + 1, Tp + 2, \ldots, 2013$.

**Line 125–126:** First, the averaged WY before Tp ($\overline{WY_{ob}}$, horizontal black line in Figure 2b) is defined as:

$$\overline{WY_{ob}} = \frac{\sum_{t=1982}^{t=Tp} WY_o(t)}{Tp - 1982 + 1} \tag{1}$$

**Line 127–129:** Next, the total WY deviation ($\Delta WY_s(t)$, black diamond in Figure 2c) can be calculated as the difference between WY observations after Tp ($WY_o(t)$, red point in Figure 2b) and the averaged WY before Tp ($\overline{WY_{ob}}$), as follows:

$$\Delta WY_s(t) = WY_o(t) - \overline{WY_{ob}}, \quad t = Tp + 1, Tp + 2, \ldots, 2013 \tag{2}$$

We apologize for not describing $WY_c(t)$ and $WY_v(t)$ clearly, and we thank the editor for pointing this out. Here, $WY_c(t)$ ($WY_v(t)$) is calculated by using the cumulative eP (LAI) values after the Tp as input into the linear regression that is built based on the cumulative data of WY and eP (LAI) before Tp. We have rephrased this description in main text.

Finally, we have rephrased these equations and related descriptions to match the six basins with different turning points, including (1) keeping "Tp" in the entire text and equations, and (2) giving the value of Tp in an example (2).

**Comment 0.7:** Section 4.2: Uncertainty in LAI has not been discussed. Furthermore, you should try to determine the sensitivity of your results to uncertain input data (in precipitation, evapo-transpiration and LAI).

**Reply:** Thanks for your suggestions. We have added a discussion on the uncertainties of GIMMS LAI used in the revised manuscript.

**Line 264–268:** GIMMS LAI has the advantage of capturing ecosystem structure, and thus is widely used to assess vegetation conditions and their effects on hydrological changes (Zhu et al., 2016; Forzieri et al., 2020; Gonsamo et al., 2021). While, LAI ignores vegetation's physiology process (Fang et al., 2019); Hu et al. (2022) indicates that LAI can cope with hydro-climatic fluctuations in arid environments, while the tradeoff between ecosystem structure (LAI) and physiology (photosynthesis per unit leaf area) becomes stronger in humid climates. Thus, using LAI products in energy-limited regions may result in biased assessments of vegetation effects on water yield.

Thanks for your suggestions about the uncertainties. We acknowledge that it is very important to use multi-source climate and vegetation data. This can provide a clear visualization about uncertainties among datasets and associated results. However, our study here focuses on a vast and mountainous watershed, where the observation networks and related datasets are limited and also are extremely difficult to acquire. In addition, popular climate data, e.g. MSWEP, ERA5, CRU, do not perform well in such complex environments and limited ground observations (Sun and Su,

2020). To overcome this, we collected the long-term annual runoff observations to indicate water yield changes. And, the high-resolution precipitation data is developed for depicting spatial and temporal precipitation changes in the Yarlung Zangbo River basin (UBR basin in this study) by Sun and Su 2020. Hence, the dataset about precipitation may be the best for the UBR basin, to our knowledge. For evaporation, GLEAM dataset has been widely validated across world, and particularly in various hydroclimates in China. There are some uncertainties among different LAI datasets due to input data, retrieval algorithms and the vegetation characterization, but GIMMS LAI3g is used in this study according to previous studies that have analyzed the effects of vegetation greenness on water yield pattern (Zhu et al., 2016; Forzieri et al., 2020; Gonsamo et al., 2021). Thus, this study provides the essential information for water resource management in the UBR basin that received less attention before based on the multi-station runoff observations and climate gridded data.

**Comment 0.8:** L272-275: Based on Fig. S8, I do not clearly see the "peak water". Please revise the text and/or the figure to support your statement.

**Reply:** Thanks for your questions. We here discuss our results and "peak water" using the framework (`https://www.nature.com/articles/s41558-017-0049-x/figures/1,` Huss and Hock, 2018) according to the second reviewer in the first round of review. Also, we provide the reasons to support our statement in the revised manuscript.

**Line 283–286:** In addition, in the headwaters, WY resulting from the cryosphere continues to increase with temperature until a maximum is reached, beyond which cryospheric contribution to total WY begins to decrease (Figure S8a), which may show that the melt waters from glaciers have already surpassed the "peak water". The hydrological changes will substantially affect future water resources management, and thus the projections of the occurrence time of "peak water" will be important in managing mountainous water resources.

**Comment 0.9:** Figure S3: Please explain what the error bars indicate.

**Reply:** Thanks for your comments. We have revised the definitions of the error bars for Figure S6.

**References**

Bibi, S., Wang, L., Li, X., Zhou, J., Chen, D., and Yao, T.: Climatic and associated cryospheric, biospheric, and hydrological changes on the Tibetan Plateau: A review, International Journal of Climatology, 38, e1–e17, 2018.

Bonferroni, C. E.: Il calcolo delle assicurazioni su gruppi di teste, Studi in onore del professore salvatore ortu carboni, pp. 13–60, 1935.

Fang, H., Baret, F., Plummer, S., and Schaepman-Strub, G.: An overview of global leaf area index (LAI): Methods, products, validation, and applications, Reviews of Geophysics, 57, 739–799, 2019.

Forzieri, G., Miralles, D. G., Ciais, P., Alkama, R., Ryu, Y., Duveiller, G., Zhang, K., Robertson, E., Kautz, M., Martens, B., et al.: Increased control of vegetation on global terrestrial energy fluxes, Nature Climate Change, 10, 356–362, 2020.

[Figure]

Figure S6. The relationship between precipitation (P) and water yield (WY) in the entire UBR basin. **The error bar represents one standard deviation of P (x-axis) and WY (y-axis).** The shading area indicates the 95% confidence interval of the fitting.

Gonsamo, A., Ciais, P., Miralles, D. G., Sitch, S., Dorigo, W., Lombardozzi, D., Friedlingstein, P., Nabel, J. E., Goll, D. S., O'Sullivan, M., et al.: Greening drylands despite warming consistent with carbon dioxide fertilization effect, Global Change Biology, 27, 3336–3349, 2021.

Hu, Z., Piao, S., Knapp, A. K., Wang, X., Peng, S., Yuan, W., Running, S., Mao, J., Shi, X., Ciais, P., et al.: Decoupling of greenness and gross primary productivity as aridity decreases, Remote Sensing of Environment, 279, 113 120, 2022.

Huss, M. and Hock, R.: Global-scale hydrological response to future glacier mass loss, Nature Climate Change, 8, 135–140, 2018.

Pellicciotti, F., Buergi, C., Immerzeel, W. W., Konz, M., and Shrestha, A. B.: Challenges and uncertainties in hydrological modeling of remote Hindu Kush–Karakoram–Himalayan (HKH) basins: suggestions for calibration strategies, Mountain Research and Development, 32, 39–50, 2012.

Sun, H. and Su, F.: Precipitation correction and reconstruction for streamflow simulation based on 262 rain gauges in the upper Brahmaputra of southern Tibetan Plateau, Journal of Hydrology, 590, 125 484, 2020.

Wang, L., Cuo, L., Luo, D., Su, F., Ye, Q., Yao, T., Zhou, J., Li, X., Li, N., Sun, H., et al.: Observing multi-sphere hydrological changes in the largest river basin of the Tibetan Plateau, Bulletin of the American Meteorological Society, 2022.

Yang, X., Yong, B., Ren, L., Zhang, Y., and Long, D.: Multi-scale validation of GLEAM evapotranspiration products over China via ChinaFLUX ET measurements, International Journal of Remote Sensing, 38, 5688–5709, 2017.

Yao, T., Xue, Y., Chen, D., Chen, F., Thompson, L., Cui, P., Koike, T., Lau, W. K.-M., Lettenmaier, D., Mosbrugger, V., et al.: Recent third pole's rapid warming accompanies cryospheric melt and water cycle intensification and interactions between monsoon and environment: Multidisciplinary approach with observations, modeling, and analysis, Bulletin of the American Meteorological Society, 100, 423–444, 2019.

Zhu, Z., Piao, S., Myneni, R. B., Huang, M., Zeng, Z., Canadell, J. G., Ciais, P., Sitch, S., Friedlingstein, P., Arneth, A., et al.: Greening of the Earth and its drivers, Nature climate change, 6, 791–795, 2016.

**Reviewer 1**

**Main Comments**

**Comment 1.1:** I appreciate the effort that authors put into revising their manuscript and I think it has improved considerably. However, I still have some problems with it. This is mainly concerned with your data and statistics. For example you are writing: "However, with the help of the Second Tibetan Plateau Scientific Expedition and Research, the observation networks in meteorology, cryosphere and hydrology will be built, which is expected to benefit reliable precipitation and evaporation estimation, and make developing physically-based cryosphere-hydrological modeling possible". This sounds to me like you cannot really estimate your uncertainty right now and it is unclear if your analysis would give you the same results if you could rerun your analysis with better data.

**Reply:** Thanks for your comments. Our study focuses on several watersheds in the UBR basin and investigate the hydrological changes and potential causes based on observed runoff data and new precipitation product developed for this region. While, the study scale in space and time may ignore some important process that are much more significant in smaller scales. Thus, we expect that more observation networks and experimental datasets in the field scale will help us understand the physical mechanisms of the hydrological cycle in this region. We made some revisions as following:

**Line 271–274:** With the help of the Second Tibetan Plateau Scientific Expedition and Research, the observation networks in meteorology, cryosphere and hydrology will be built, which is expected to better understand hydrological process in this region, and make developing physically-based cryosphere-hydrological modeling possible (Wang et al., 2022).

**Comment 1.2:** Still, this not really your fault that better data does not exist (yet?), so it seems like your analysis is the best you could do given the current data constrains. This should also be emphasized in the paper.

**Reply:** Thanks for your understanding. To my knowledge, this study uses the multi-station observed runoff and new precipitation product designed for the UBR basin, and thus provides some important information that previous studies did not reveal. We have emphasized it in main text.

**Comment 1.3:** In addition, this makes it much more important that your work is reproducible, which it is not at the moment. Neither the code nor the data is openly available. This means even if better data becomes available your results still cannot be verified. Therefore, to make this paper a valuable contribution I think it is important that a documented and citable repository exists for the code (preferably as Jupyter Notebooks) and if possible, the data as well.

**Reply:** We agree with the reviewer. Reproducibility is ensured through GitHub. We have provided a link for accessing the Jupyter Notebook via the git repository (`https://github.ugent.be/haohaoli/HESS-2022-Water-Yield.git`). However, runoff observations can't be shared due to a confidentiality contract.

**Comment 1.4:** I also think you misunderstood my comment 1.14. With "corrected" my question was if you account for doing multiple statistical tests in a row (`https://en.wikipedia.`

org/wiki/Multiple_comparisons_problem). If not you have to do a correction (e.g. https://en.wikipedia.org/wiki/Bonferroni_correction).

**Reply:** Thanks for pointing it out. We use the Bonferroni method to correct p values that still support the main results, and report the method and relevant descriptions using the corrected p values in the revised manuscript. But, the change from the significant to non-significant correlation using the corrected p values is observed in the relationship between vegetation contributions and total water yield changes in the UYZR basin (see the updated Figure 6b).

**Line 151–152:** The Student's t-test with the Bonferroni correction (Bonferroni, 1935) is used to detect the statistical significance of the correlation coefficient at the level of 0.05.

**Minor comments**

**Comment 1.5:** The figure caption of Figure 2 isn't really that helpful. First, this is an example and not a schema. Secondly, It does not really "show" me how the effects are estimated. This would need additional explanations in the figure and the caption to become a schema.

**Reply:** Thanks for your questions. We describe the DMC used in this study by (1) providing a general framework using equations and relevant statements, and (2) giving an example to help describe the framework. Thus, in the revised version, we have rephrased the **equations** and **related descriptions** to show how to estimate the effects, while the Figure 2 gives an **example** to help us better describe these equations, where we give the value of the turning point in the caption.

**Comment 1.6:** Line 180-183: How can they be additive and offsetting at the same time?

**Reply:** Thanks for your question. The additive or offsetting effects are seen in **different basins** (Figure 5); for example, the additive effect is observed in MYZR basin, while the offsetting effect is found in HYZR basin. Also, we revised the statement in the main text.

**Line 182–184:** Climate and cryosphere – two important factors affecting WY – together contribute to over 80% average magnitude increases of WY in the entire UBR basin; however, they play both additive or offsetting roles **in different basins** (Figure 5) …

**Reviewer 2**

**Comment 2.1:** The study of Li et al. is about the analysis of long -term water yield from the Upper Brahmaputra River basin associated with potential drivers such as climate, vegetation, and the cryosphere. Based on hydro-climatic data, the authors identify turnings points to infer major water yield shifts in the basins. Furthermore, the study reports that climate may act as driver for downstream water yield while meltwaters control water yield further upstream. It is also found that melt water dynamics balance out low flow conditions induced by climate. The insights gained in this study may help to better understand large scale drivers of water yield and improve water management strategies. Besides, only minor comments regard typing errors and the language quality (see specific comments below). The manuscript generally is of good English.

**Reply:** We thank the reviewer for the constructive comments. We revised some typos in the revised version and gave a point-to-point response below.

**Specific comments**

**Comment 2.2:** p. 1, l. 4: Please modify 'find that'

**Reply:** Thanks. We have revised this.

**Comment 2.3:** p. 1, l. 5: At this stage the reader will not understand the upstream and down-stream direction of water yield while it becomes clearer after reading the entire manuscript. I recommend mentioning instead only controls of water yield in the upper and lower reaches.

**Reply:** Thanks. We now realized that the description may confuse readers, and thus we revised the related descriptions. **Line 5–6:** ... magnitude increases in WY range from ∼10% to ∼80%,

while its directions reverse from **the increase** to **decrease** after the late 1990s.

**Comment 2.4:** p. 1, l. 14: It would be great to define water yield in this section, simply saying that it is based on the total runoff of a basin.

**Reply:** Thanks. We have used "runoff depth" to further explain "water yield" in the beginning of the introduction.

**Line 14–15:** Water yield (or runoff depth, WY) in mountainous watersheds is crucial for sustaining fragile ecosystems in headwaters ...

**Comment 2.5:** p. 2, l. 48: Please rephrase this sentence.

**Reply:** Thanks, we have rewritten this statement. **Line 48–49:** While, long-term observed runoff

records and recent high-resolution precipitation data in the UBR basin provide a valuable oppor-tunity to estimate runoff responses to warming by statistical methods.

**Comment 2.6:** p. 5, l. 85: showed instead of shown

**Reply:** Thanks, we have revised that. **Line 85:** GLEAM evaporation has been validated in different

biome types in China and **showed** high correlations with in-situ eddy covariance E (Yang et al., 2017).

**Comment 2.7:** p. 5, l. 100: In this line, you should add further explanation on turning points, their definitions and shortly how the Pettitt method works.

**Reply:** Thanks for your suggestions. We have added the advantages of the Pettitt method. **Line 101–102:** The Pettitt method has the advantages of the non-parametric, rank-based and

distribution-free for finding the abrupt variation in a time series.

**Comment 2.8:** p. 5, l. 106: The fact that the cryosphere represents meltwaters from snow and ice should appear earlier in the text.

**Reply:** Thanks. We have provided some descriptions in the last paragraph in Introduction.

**Line 52–55:** And then, we use DMC method to estimate the effects of climate (effective precipitation, P-E, eP), vegetation (Leaf Area Index, LAI), and cryosphere (e.g. melt waters from glaciers and snow) on magnitude and direction changes in WY (See Data and Methods).

**Comment 2.9:** p. 8, l. 158: The first sentence is redundant to what is said earlier in the methods.

**Reply:** Thanks, we have deleted this sentence.

**Comment 2.10:** p. 9, l. 168: becomes instead of become

**Reply:** Thanks, we have revised the typo. **Line 170–171:** ... we find that **the trend is** positive

before Tp, but **becomes** negative afterward in most basins.

**Comment 2.11:** p. 11, l. 205: a word is missing as it seems. Maybe observation, result, or finding?

**Reply:** I am sorry for that. We have replaced "The is ..." with "This is ..." **Line 206–207:** ...

and snow under continuous warming. **This is** further verified by the relationship of cryospheric contributions ...

**Comment 2.12:** p. 13, l. 225: controlling instead of control

**Reply:** Thanks, we have revised the whole sentence. **Line 228–229:** Climate, especially precipitation, still **controls** the declining WY trend after ...

**Comment 2.13:** p. 13, l. 242-244: Please rephrase this sentence.

**Reply:** Thanks for your comments. We have described the sentence clearly. **Line 245–247:** While,

in some other mountainous basins, human activities, such as urbanization, dam regulation and irrigation, may consume severely water resources or change seasonal runoff patterns; **thus it is**

**necessary to** consider anthropogenic impacts when assess river flow changes via the statistical models **in these regions**.

**Comment 2.14:** p. 13, l. 244: "on the one hand" is missing previously

**Reply:** Thanks. "In addition" may be better to link these sentences.

**Comment 2.15:** p. 13, l. 245: Please add further details on the nonlinear processes you are referring to here.

**Reply:** Thanks. We have added some examples into the main text. **Line 248–250:** ... some nonlinear process among water, vegetation and cryospheric melting. For example, the role of vegetation in hydrological process is expected to be complex due to biophysical (e.g., via transpiration and albedo changes) and biochemical (e.g., via $CO_2$ uptake and release) feedbacks (Bonan 2008; Krich et al. 2022).

**Comment 2.16:** p. 13, l. 246: 'interactions of'

**Reply:** Thanks. We have rephrased this sentence.

**Comment 2.17:** p. 14, l. 251: remove 'while'

**Reply:** We have removed "while".

**Comment 2.18:** p. 15, l. 283: remove 'with'

**Reply:** We have removed "with".

**Comment 2.19:** Table 1: Correct 'turning point'.

**Reply:** Thanks for your comments. We have kept the abbreviation "Tp" of "turning point" in the revised manuscript.

**Comment 2.20:** Figure 4: Labels are too small, please enlarge for better visibility.

**Reply:** Thanks. We have adjusted the font size for all figures in main text and supplementary.

**Comment 2.21:** Figure S8: It is highly questionable why, for some data clouds (subplot b,d,f), the polynomial fitting was used. Generally, the best fit among several methods is reported, except there is an important reason for a specific method to give.

**Reply:** Thanks for your questions. We tried to link our results with present findings, e.g. "peak water" according to the second reviewer in the first round of review. Hence, we referred to this framework (`https://www.nature.com/articles/s41558-017-0049-x/figures/1`, Huss and Hock, 2018) that shows the response of runoff from a glacierized basin to continuous atmospheric warming (years). Hence, we used the polynomial curve to depict relationship between the cryospheric contribution (this study) and the air temperature. Indeed, we can see the clear pattern in headwaters that cryospheric contribution increases firstly and decrease later with temperatures, which may imply the "peak water" as shown in (Huss and Hock, 2018). Honestly, we acknowledge that the simple statistical methods, e.g. polynomial curve, may be not the best way to describe the relationship between cryospheric contribution (y-axis) and air temperature (x-axis) or revealing the "peak water". Thus, we keep cautions about the explanation, and only report that the HYZR basin may have surpassed the "peak water" in the revised text.

**References**

Bonan, G. B.: Forests and climate change: forcings, feedbacks, and the climate benefits of forests, science, 320, 1444–1449, 2008.

Huss, M. and Hock, R.: Global-scale hydrological response to future glacier mass loss, Nature Climate Change, 8, 135–140, 2018.

Krich, C., Mahecha, M. D., Migliavacca, M., De Kauwe, M. G., Griebel, A., Runge, J., and Miralles, D. G.: Decoupling between ecosystem photosynthesis and transpiration: a last resort against overheating, Environmental Research Letters, 17, 044 013, 2022.

---

## Author Response (AR3)

**Response to Editors' and Reviewers' Comments: **Significant regime shifts of historical water yield in the Upper Brahmaputra River basin**

Dear Dr. Giulia Zuecco,

Thanks for the comments and suggestions from you and three reviewers in the past several rounds of review. Now, we addressed some technological corrections, mainly including (1) careful English checks, (2) updated figures, and (3) a repository for data, Python codes and LaTex scripts. We hope that the revised manuscript can meet the standards of the Journal *Hydrology and Earth System Sciences*.

In addition, we decided to use a much more clear, tightly constructed title **"Significant regime shifts of historical water yield in the Upper Brahmaputra River basin"**. We believe that it can improve the chances of the manuscript being discovered by relevant researchers and the public.

Below we provide a point-by-point response to the comments and concerns raised by the editor and reviewers. All modifications in the manuscript have been marked.

Sincerely

Hao Li (on behalf of all coauthors)
PhD candidate, Hao.liwork@ugent.be
Hydro-Climate Extremes Lab, Ghent University
Nov 12, 2022

Li Hao

**Editor**

**Comment 0.1:** Thank you for submitting a revised version of the manuscript. As you can see, the first reviewer appreciated the revised manuscript, but he noticed that the Python scripts cannot be accessed because a password is required.

**Reply:** We are so sorry that we did not double check that. We have created a Git repository that is connected Zenodo (`https://doi.org/10.5281/zenodo.7315564`).

**Comment 0.2:** A careful revision of the English by a native speaker is needed due to unclear sentences, typos, and repeated words (e.g., lines 24-26 in the manuscript without track changes, lines 246-247, lines 266-268 etc.).

**Reply:** Thank you very much for your suggestion. We have revised the whole manuscript and carefully proofread the manuscript to minimize typographical, grammatical, and bibliographical errors. A native English speaker was also invited to check the language. We believe that the language is now acceptable for the review process.

Relevant revisions are shown:

**Line 22–25:** Therefore, WY changes over this region significantly affect water availability, terrestrial and aquatic ecosystems which are vital for sustaining the livelihoods of approximately two billion people (Immerzeel et al., 2010). Despite some in-situ observations and estimates from state-of-the-art remote sensing (Wang et al., 2021), total river runoff has never been reliably assessed in this region, and its response to global warming remains unclear.

**Line 245–248:** However, in some mountainous basins, human activities such as urbanization, dam regulation and irrigation may consume a significant portion of water resources or induce changes in seasonal runoff patterns. Therefore, it is necessary to account for anthropogenic impacts when assessing river flow changes via the statistical models in these regions, in addition to the factors considered in this study.

**Line 267–270:** However, LAI ignores the vegetation's physiological process (Fang et al., 2019). Hu et al. (2022) have indicated that LAI can cope with hydro-climatic fluctuations in arid environments, but the tradeoff between ecosystem structure (LAI) and physiology (photosynthesis per unit leaf area) is stronger in humid climates. Thus, using LAI products in energy-limited regions may result in some biased assessments of vegetation effects on water yield.

**Comment 0.3:** Fig. 2a: Please add the legend for the dots.

**Reply:** Thanks for your comment. We have updated Figure 2a with a clearer illustration.

**Comment 0.4:** Please check that all acronyms are explained and described correctly in the manuscript.

**Reply:** Thanks. We have checked the acronyms in the manuscript. Now, all of them have been defined in a clear way.

**Comment 0.5:** Fig. 4a: Please explain in the caption what $\triangle R$ means.

[Figure]

Figure 4. Water yield regime shifts in the entire UBR basin. (a) Magnitude of water yield changes. The black "x" signal shows the mean of water yield, **and the relative change is labeled in number (%) in each boxplot**. (b) Direction of water yield changes. The black hatching represents the statistically significant trend (p < 0.05). The color of boxes represents before (light color) and after (dark color) Tp period.

**Reply:** Thanks for your comment. We deleted $\Delta$R in the figure. Instead, we added detailed descriptions in the caption.

**Comment 0.6:** Fig. S1, S2, S5, S7 and S9: Please increase the size of the labels (some are too small and it is difficult to read them).

**Reply:** Thanks for your reminder. We have updated these figures in the supplementary.

**References**

Fang, H., Baret, F., Plummer, S., and Schaepman-Strub, G.: An overview of global leaf area index (LAI): Methods, products, validation, and applications, Reviews of Geophysics, 57, 739–799, 2019.

Hu, Z., Piao, S., Knapp, A. K., Wang, X., Peng, S., Yuan, W., Running, S., Mao, J., Shi, X., Ciais, P., et al.: Decoupling of greenness and gross primary productivity as aridity decreases, Remote Sensing of Environment, 279, 113 120, 2022.

Immerzeel, W. W., Van Beek, L. P., and Bierkens, M. F.: Climate change will affect the Asian water towers, Science, 328, 1382–1385, 2010.

Wang, L., Yao, T., Chai, C., Cuo, L., Su, F., Zhang, F., Yao, Z., Zhang, Y., Li, X., Qi, J., et al.: TP-River: Monitoring and quantifying total river runoff from the Third Pole, Bulletin of the American Meteorological Society, 102, E948–E965, 2021.

**Reviewer 1**

**Technical corrections**

**Comment 1.1:** Thank you again for addressing my comments. I think the paper is now in a state that can be published. There is just one problem that I cannot access the repository, as I asks me for a password, which I do not have. I would be better to just publish it as a public repository on Github, while also making citable with Zenodo (https://zenodo.org). This would make sure that a fixed version of the Code is assigned to the paper.

**Reply:** Thank you very much for your comments and suggestions in past several rounds of review. We have created a Git repository that is connected by Zenodo (`https://doi.org/10.5281/zenodo.7315564`).

Also, we decided to use the new title **"Significant regime shifts of historical water yield in the Upper Brahmaputra River basin"**. We believe that it is much more clear, tightly constructed and also can improve the chances of the manuscript being discovered by relevant researchers and the public. We are very glad to hear your opinions.